# Sustained rhoptry docking and discharge requires *Toxoplasma gondii* intraconoidal microtubule-associated proteins

Nicolas Dos Santos Pacheco [1,5], Albert Tell i Puig [1,5], Amandine Guérin [2,5], Matthew Martinez [3], Bohumil Maco [1], Nicolò Tosetti[1], Estefanía Delgado-Betancourt [1], Matteo Lunghi [1], Boris Striepen [2], Yi-Wei Chang [3,4] ✉ & Dominique Soldati-Favre [1] ✉

In Apicomplexa, rhoptry discharge is essential for invasion and involves an apical vesicle (AV) docking one or two rhoptries to a macromolecular secretory apparatus. *Toxoplasma gondii* is armed with 10–12 rhoptries and 5-6 microtubule-associated vesicles (MVs) presumably for iterative rhoptry discharge. Here, we have addressed the localization and functional significance of two intraconoidal microtubule (ICMT)-associated proteins instrumental for invasion. Mechanistically, depletion of ICMAP2 leads to a dissociation of the ICMTs, their detachment from the conoid and dispersion of MVs and rhoptries. ICMAP3 exists in two isoforms that contribute to the control of the ICMTs length and the docking of the two rhoptries at the AV, respectively. This study illuminates the central role ICMTs play in scaffolding the discharge of multiple rhoptries. This process is instrumental for virulence in the mouse model of infection and in addition promotes sterile protection against *T. gondii* via the release of key effectors inducing immunity.

Microtubules are dynamic polymers made of tubulins. These components of the eukaryotic cell cytoskeleton contribute to essential cellular functions such as motility, shape maintenance, division, and cargo transport[1]. *Toxoplasma gondii* is a unicellular eukaryotic parasite belonging to the phylum of Apicomplexa that is unified by the presence of an apical complex, composed of specialized secretory organelles and unique tubulin-based cytoskeletal structures (Fig. 1a)[2,3]. The cytoskeleton of the parasite is made of 22 subpellicular microtubules (SPMTs) that emerge from the apical polar ring (APR), which presumably serves as the microtubule-organizing center[4–6]. The SPMTs form a corset spanning two-thirds of the parasite length and, unlike typical eukaryotic microtubules, are remarkably stable and resistant to cold treatment or detergents[7]. Situated apically, the conoid is a dynamic organelle able to extrude through the APR in activated extracellular parasites[8,9]. The cone of the conoid is composed of atypical tubulin fibers of 9 protofilaments organized in a distinctive comma shape, contrasting with the hollow-tube organization of 13 protofilaments seen in canonical microtubules[10,11]. The conoid is topped by proteinaceous preconoidal rings (PCRs) recently shown to serve as a hub for essential motility and invasion factors[12,13]. Inside the conoid are two short microtubules of unknown function referred to as intraconoidal microtubules (ICMTs). The intraconoidal microtubule-associated protein 1 (ICMAP1) is the only protein described to date to be exclusively associated with the ICMTs[14].

Two sets of apical secretory organelles, the micronemes and rhoptries, are in close vicinity of the conoid. The numerous small,

[1]Department of Microbiology and Molecular Medicine, Faculty of Medicine, University of Geneva, Geneva, Switzerland. [2]Department of Pathobiology, School of Veterinary Medicine, University of Pennsylvania, Philadelphia, PA, USA. [3]Department of Biochemistry and Biophysics, Perelman School of Medicine, University of Pennsylvania, Philadelphia, PA, USA. [4]Institute of Structural Biology, Perelman School of Medicine, University of Pennsylvania, Philadelphia, PA, USA. [5]These authors contributed equally: Nicolas Dos Santos Pacheco, Albert Tell i Puig, Amandine Guérin. ✉e-mail: ywc@pennmedicine.upenn.edu; dominique.soldati-favre@unige.ch

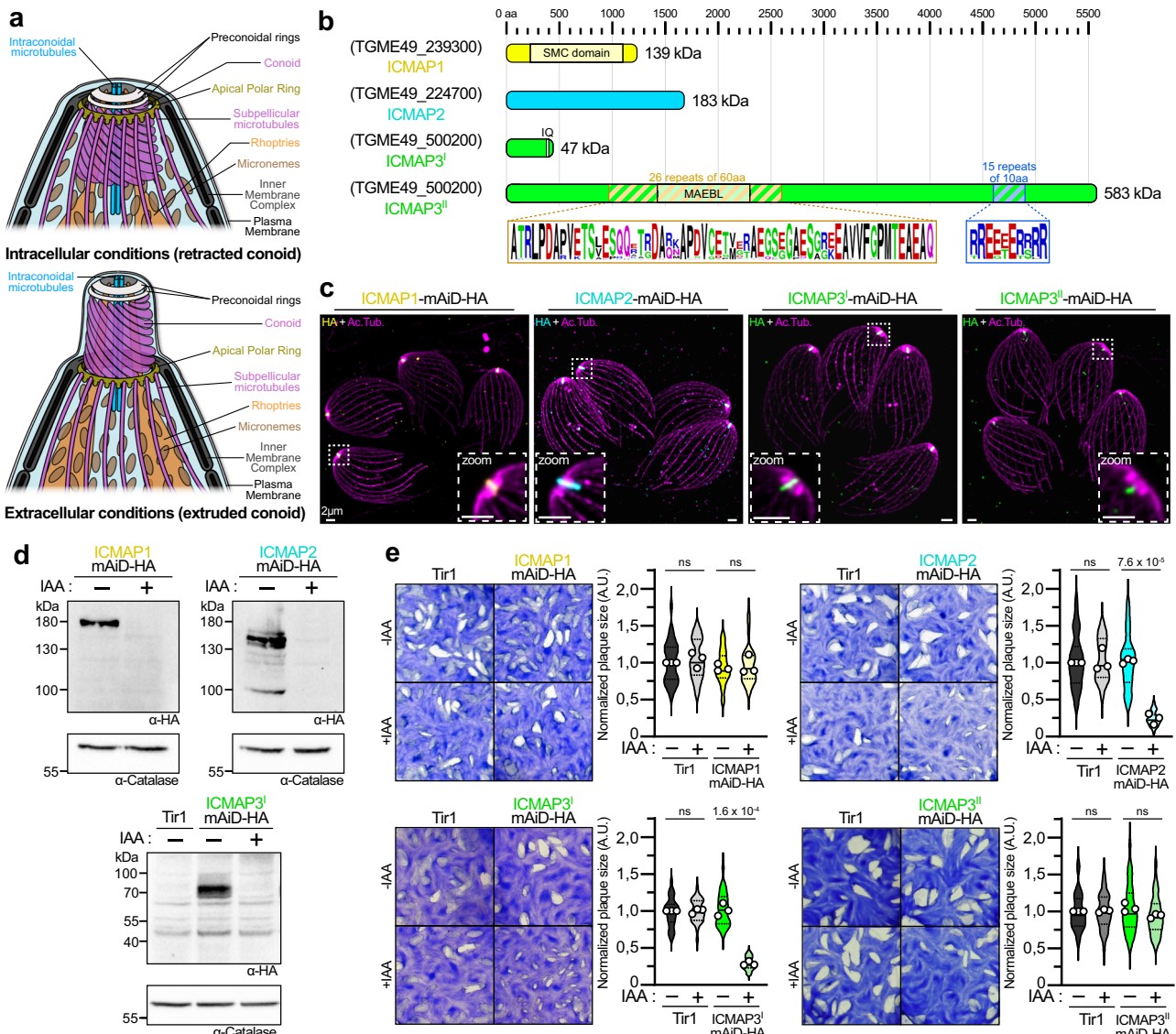

**Fig. 1 | Characterization of four intraconoidal microtubule proteins.**
**a** Schematic of *T. gondii* apical complex in retracted or extruded states. Modified from ref. 2. **b** Schematic representation of the four proteins of interest and their domains. **c** Localization of ICMAP1-, ICMAP2-, ICMAP3$^I$- and ICMAP3$^{II}$-mAiD-HA by U-ExM in intracellular parasites. For each, a zoom insert focused on the apical region is shown. The zoom insert has an adjusted contrast compared to the full picture. Ac.Tub, acetylated tubulin. **d** Downregulation of ICMAP1-, ICMAP2- and ICMAP3$^I$-mAiD-HA by western blot. Catalase is used as loading control. Scale in

kDa. **e** Fitness of ICMAP1-, ICMAP2-, ICMAP3$^I$- and ICMAP3$^{II}$-mAiD-HA strains assessed by plaque assay. Tir1 is the parental control strain. Quantification of $n = 3$ independent biological triplicate is presented on the right and representative pictures on the left. Mean of the individual replicates are presented as white dots, the median of the total distribution of plaque size is presented as a solid line while the first and third quartiles are presented as dotted lines. Unpaired two-tailed Student's *t*-tests were performed where ns if $P > 5 \times 10^{-2}$.

rod-shaped micronemes secrete sets of perforins, adhesins and proteases that are instrumental for egress, motility, host cell attachment and invasion[15]. Upon calcium and phosphatidic acid-dependent stimulation, micronemes funnel through the conoid to release their content at the apical tip of the parasite[16]. On the other hand, rhoptries are elongated club-shaped organelles containing essential proteins for invasion and effectors implicated in subversion of the host cellular functions[17–19]. Unlike *Cryptosporidium* and *Plasmodium* species that possess one and two rhoptries respectively, *T. gondii* exhibits 10–12 rhoptries usually arranged in a bunch, with two of them extending their neck through the conoid, making contact with an elaborate rhoptry secretory apparatus (RSA) for discharge[20–22]. Proteins that compose the RSA are conserved in other Alveolates species and involve, among others, non-discharge proteins such as Nd6 or Nd9, and cysteine repeats modular proteins (CRMPs)[21,23]. Remarkably, *T.*

*gondii* harbors multiple rhoptries and hence is capable of not only discharging rhoptries into cells it invades but also injecting other cells[24] offering a novel mechanism to manipulate the host environment, which may be particularly important in the brain, where this neurotropic parasite encysts and persists[25].

Recently, cryo-electron microscopy (cryo-EM) and tomography (cryo-ET) were used to decipher the fine architecture of *T. gondii* apical complex[26]. The organization and composition of the conoid tubulin fibers and SPMTs were described at an unprecedented level of resolution[27–30]. Specifically, cryo-ET unraveled details of the machinery involved in rhoptry discharge[20,23,31,32]. The necks of the two most apical rhoptries were seen near the ICMTs, inside the conoid, and docking to an apical vesicle (AV), which in turn was docked to the parasite plasma membrane. Importantly, 4–5 microtubule-associated vesicles (MVs) were found to line up parallel to the ICMTs. These MVs

are presumably involved in the reloading of the AV, enabling several rounds of rhoptry exocytosis, prior to successful invasion[24]. Finally, along one of the two ICMTs, an enigmatic fibrous material was described but its origin, composition and function are still unknown[20]. Pioneering EM work[4], recently further supported by cryo-ET[20], placed the ICMTs at the center of the rhoptry secretion machinery, suggesting a key role in the positioning and discharge of these organelles. However, the molecular composition and function of the ICMTs has been largely unexplored to date.

Here, along with the previously described ICMAP1, we identified and functionally characterized three additional ICMT proteins, ICMAP2, ICMAP3[I] and ICMAP3[II]. We used Ultrastructure Expansion Microscopy (U-ExM) to assign individual ICMAPs to sub-compartments of the ICMTs, and cryo-ET to resolve cellular structural consequences of lacking each of the ICMAPs. Together, this study illuminates the central structural role ICMTs play in iterative rhoptry discharge in *T. gondii*, a process crucial to virulence and modulation of the host immune response.

## Results

### Identification of two fitness-conferring ICMT-associated proteins

To unravel the function of ICMTs, we scrutinized the "apical complex" cluster of the Hyperplexed Localization of Organelle Proteins by Isotope Tagging (hyperLOPIT) dataset[33] for candidate ICMT-resident proteins and prioritized those conferring fitness as deduced from the genome-wide CRISPR-Cas9 screen[34]. Given that ICMTs are exclusively found in the coccidian subgroup of Apicomplexa[4,35], we filtered for proteins present in coccidia but absent in *Plasmodium* and *Cryptosporidium spp*. TGME49_224700 fulfilled all of these criteria and was investigated along with TGME49_500200, which was shown previously to localize to an elongated apical structure reminiscent of the ICMTs[36] (Supplementary Fig. 1a). Of note, the TGME49_500200 locus, known in previous ToxoDB versions as TGME49_285140-285150, forms a single ORF coding for two protein variants through alternative splicing (see Supplementary Discussion and Supplementary Fig. 2). While the SMC (Structural Maintenance of Chromosome) domain of ICMAP1 was already described[14], no recognizable domain could be identified for TGME49_224700. A putative IQ motif could be identified for the short isoform of TGME49_500200, while the long isoform of the gene shows weak homology to a MAEBL (Merozoite Adhesive Erythrocyte Binding protein) domain and two large regions containing repeated sequences of unknown function (Fig. 1b).

The three candidate proteins as well as ICMAP1[14] were modified to append a C-terminal mini-auxin-induced degron cassette (mAiD) and hemagglutinin tags (HA) enabling conditional protein ablation and localization, respectively (Supplementary Fig. 1b)[37,38]. By U-ExM, ICMAP1, TGME49_224700 and the short isoform of TGME49_500200 displayed a linear signal inside the conoid reminiscent of the ICMTs. Hence, TGME49_224700 was named ICMAP2, while the short isoform of TGME49_500200 was named ICMAP3[I] (Fig. 1c). The long isoform of TGME49_500200 localized to a small punctum in the basal portion of the ICMTs and was named ICMAP3[II]. Fractionation of parasite lysates showed that while ICMAP1 and ICMAP3[I] were readily soluble in PBS, ICMAP2 was mostly insoluble, even in the presence of $Na_2CO_3$ (0.1 M, pH 11.5) (Supplementary Fig. 1c). ICMAP3[II] could not be detected intact by western blot, which may be due to its high molecular weight. ICMAP1, ICMAP2 and ICMAP3[I] were already found in late-stage daughter cells during parasite replication, while ICMAP3[II] was only detectable in mature parasites (Supplementary Fig. 1d).

Upon addition of auxin (IAA) to the media, each of the four proteins could be efficiently degraded as shown by western blot analysis and indirect immunofluorescence assay (IFA) (Fig. 1d and Supplementary Fig. 1e). Parasite fitness upon depletion of the individual ICMAPs was then assessed by plaque assay (Fig. 1e). ICMAP1- and

ICMAP3[II]-depleted parasites exhibited plaque sizes comparable to the Tir1 parental strain whereas depletion of ICMAP2 or ICMAP3[I] led to a ~75% reduction in plaque size. Additionally, ICMAP1- and ICMAP3[II]-depleted parasites did not show significant invasion defects (Supplementary Fig. 1f). To confirm that ICMAP2 and ICMAP3[I] were fitness conferring but not essential for parasite survival, knock-out lines were generated (ΔICMAP2 and ΔICMAP3) and shown to form plaques smaller than the control ΔKu80 strain (Supplementary Fig. 1g, h).

### Ultrastructure expansion microscopy revealed ICMAPs localization to distinct sub-compartments of the ICMTs

To inspect the localization of ICMAPs in more depth, ICMAP1, ICMAP3[I] and ICMAP3[II] were individually fused with Ty epitopes in an ICMAP2-mAiD-HA background strain.

Using immunostaining against Ty and HA epitopes, confocal microscopy showed partial colocalization of ICMAP1 and ICMAP2 (Supplementary Fig. 3a). U-ExM revealed the ICMAP1 labeled structure to be shorter, restricted to the most apical part of the ICMTs and juxtaposed to the ICMAP2 signal in intracellular and activated extracellular parasites with extruded conoid (Fig. 2a). ICMAP1, ICMAP2 and tubulin signal intensities were plotted and revealed colocalization of ICMAP2 with the ICMTs, and a noticeable lateral shift between ICMAP1 and ICMAP2 (Fig. 2b). Despite the ICMAP1 signal not being present in all parasites of the ICMAP2-mAiD-HA background, the number of ICMAP1-bearing parasites decreased considerably (2.5-fold) upon ICMAP2 depletion (Fig. 2c). This partial loss of ICMAP1 upon ICMAP2 depletion was confirmed by western blot with a slight decrease in ICMAP1 levels (Fig. 2d).

In contrast to ICMAP1, ICMAP3[I] localization mirrored that of ICMAP2 by confocal microscopy (Supplementary Fig. 3b). By U-ExM, both in intracellular and extracellular parasites, ICMAP3[I] and ICMAP2 colocalized along the entire ICMT length (Fig. 2e). This is also evident when the fluorescent signals of ICMAP3[I], ICMAP2 and tubulin are plotted (Fig. 2f). Upon ICMAP2 depletion, ICMAP3[I] was no longer detected at the apex of the parasite, while its overall abundance, as measured by western blot, appeared unchanged (Fig. 2g, h). In contrast, ICMAP3[I] depletion did not impact the apical signal of ICMAP2 (Fig. 2i).

ICMAP3[II] and ICMAP2 did not colocalize perfectly by confocal microscopy (Supplementary Fig. 3c). By U-ExM, ICMAP3[II] was observed only in the basal region of the ICMTs (Fig. 2j). This experiment was not carried out with extracellular parasites as ICMAP3[II] required a cold-methanol fixation to be detectable by U-ExM, a protocol that preserves well the shape of intracellular parasites but induces dramatic abnormalities on extracellular ones. Like ICMAP3[I], ICMAP2 depletion led to the loss of ICMAP3[II] in all parasites (Fig. 2k), indicating that ICMAP2 is essential for the localization of both ICMAP3 isoforms.

### Depletion of ICMAP1 leads to the loss of the fibrous material associated with ICMTs

ICMT architecture and associated structures are discernable using cellular cryo-ET[26]. In wild-type *T. gondii* tachyzoites, cryo-ET shows a fibrous material associated with one of the ICMTs (called ICMT$_1$ hereafter), a line of MVs with the second ICMT (called ICMT$_2$ hereafter), and two rhoptry necks with their tips docked to the AV (Fig. 3a and Supplementary Movie 1). The U-ExM localization of ICMAP1, at the apical part of the ICMTs and slightly shifted to one side, led us to hypothesize that ICMAP1 might be linked to the fibrous material. To test this, we performed cryo-ET analysis on ICMAP1-depleted parasites (Fig. 3b and Supplementary Movie 2). Upon ICMAP1 depletion, the AV was present in all analyzed tomograms ($n = 20$) and most of them had two rhoptries docked as seen for wild-type (Fig. 3c, d). The average length of the ICMTs was slightly shorter in ICMAP1-depleted parasites ($340 \pm 41$ nm) compared to a previously reported wild-type strain ($376 \pm 59$ nm) (Fig. 3e)[20]. Most strikingly, the fibrous material usually associated with ICMT$_1$ was never observed in ICMAP1-depleted

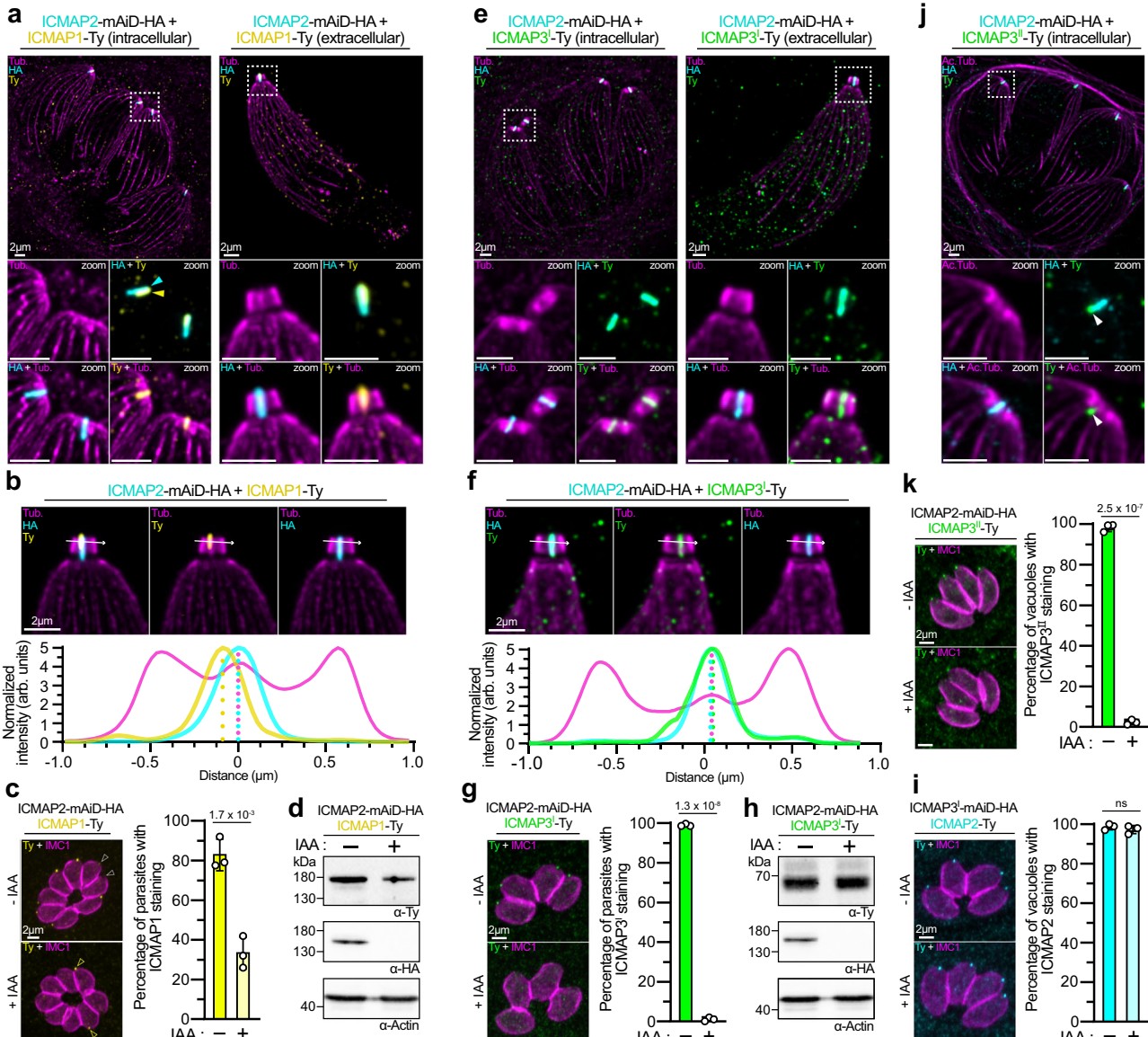

**Fig. 2 | Hierarchy and differential localization of the ICMAPs. a** ICMAP2 and ICMAP1 do not colocalize exactly when assessed by U-ExM in intracellular (left) or extracellular (right) parasites. Tub, tubulin. **b** Intensity profiles of the ICMAP1-2 signals by U-ExM on an extracellular parasite. The first and last peaks of the tubulin plots correspond to the conoid walls, while the central peak corresponds to the ICMTs. **c** ICMAP2 depletion leads to a partial loss (yellow arrowhead) of ICMAP1 signal in intracellular parasites. Note that in the ICMAP2-mAiD-HA background, ICMAP1 is not always seen at the apical pole of the parasite (white arrowhead), contrary to what is seen with the ICMAP1-mAiD-HA strain. **d** Western blot showing that the depletion of ICMAP2-mAiD-HA leads to a reduction of ICMAP1 levels. Actin is used as a loading control. **e** ICMAP3$^I$ colocalize perfectly with the ICMAP2 signal when assessed by U-ExM in intracellular (left) or extracellular (right) parasites. Ac.Tub, acetylated tubulin. Only the stacks close to the conoid were max projected

to avoid nonspecific background from above or below. **f** Intensity profiles of the ICMAP2-3$^I$ signals by U-ExM on an extracellular parasite. **g** ICMAP2 depletion leads to a complete loss of ICMAP3$^I$ signal in intracellular parasites. **h** Western blot showing that the depletion of ICMAP2-mAiD-HA does not affect the protein levels of ICMAP3$^I$. Actin is used as a loading control. **i** ICMAP3$^I$-mAiD-HA depletion does not affect the localization of ICMAP2 in intracellular parasites. **j** ICMAP3$^{II}$ colocalize only with the base of the ICMAP2 signal when assessed by U-ExM in intracellular parasites. Ac.Tub, acetylated tubulin. Only the stacks close to the conoid were max projected to avoid nonspecific background from above or below. **k** ICMAP2 depletion leads to a complete loss of ICMAP3$^{II}$ signal in intracellular parasites. For panels (**c**), (**g**), (**i**) and (**k**), $n = 3$ independent biological replicates, mean ± SD with individual replicates is presented, and unpaired two-tailed Student's $t$-tests were performed where ns if $P > 5 \times 10^{-2}$.

parasites (Fig. 3b, f). Taken together, these data demonstrate that ICMAP1 is essential for the formation or maintenance of the fibrous material, although the loss of this structure is not fitness conferring in the tested culture condition.

**Depletion of ICMAP2 or ICMAP3$^I$ leads to a defect in rhoptry discharge**

To unravel the origin of the significant fitness loss observed in ICMAP2- and ICMAP3$^I$-depleted parasites, we dissected their impact on

individual steps of the lytic cycle. First, intracellular growth assays showed that both ICMAP2- and ICMAP3$^I$-depleted parasites were able to multiply normally (Supplementary Fig. 4a). Next, gliding motility on 2D surface was assessed by trail-deposition assay[39], and no defect was detectable upon ICMAP2 or ICMAP3$^I$ depletion (Supplementary Fig. 4b). Egress was induced with an inhibitor of phosphodiesterase (BIPPO)[40] and found to be comparable between control and ICMAP2- and ICMAP3$^I$-depleted parasites (Supplementary Fig. 4c). In contrast, the invasion step was clearly affected in the mutants, albeit not fully

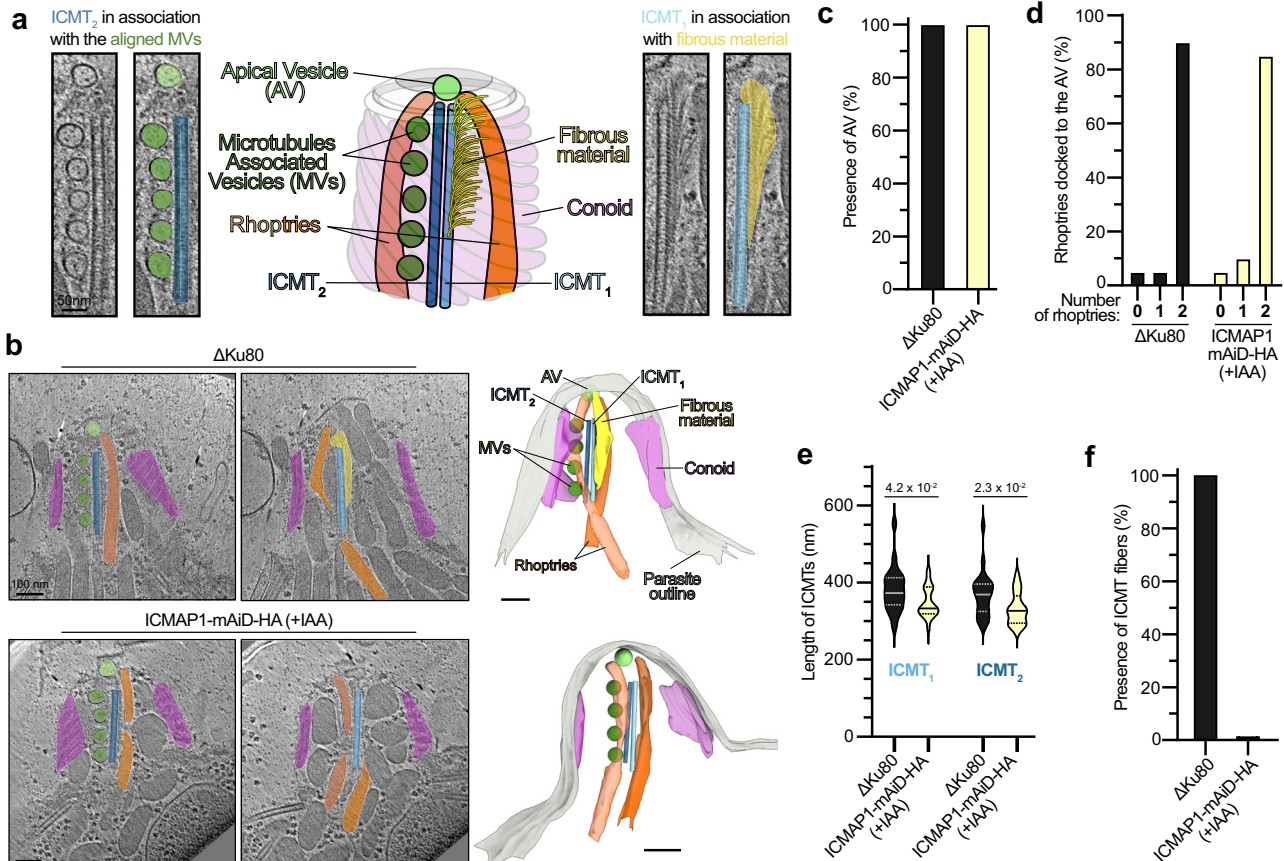

**Fig. 3 | Cryo-ET reveals that ICMAP1 is associated with the fibrous material of ICMT$_1$. a** Representative schematic and tomograms of the two ICMTs and associated structures. Aligned vesicles (MVs) are associated with ICMT$_2$, while the fibrous material is associated with ICMT$_1$. The two rhoptries inside the conoid are docked to the apical vesicle (AV). **b** Cryo-ET reveals that ICMAP1-depleted parasites lack fibrous material usually in contact with ICMT$_1$. Tomograms at two relevant cross-sections are shown on the left and the associated 3D segmentation is shown on the right. ΔKu80 is used as a control strain in all cryo-ET panels. **c** Depletion of ICMAP1 does not affect the presence of the AV. **d** Depletion of ICMAP1 does not affect the number of rhoptries docked to the AV. **e** Depletion of ICMAP1 slightly affects the length of both ICMTs. Unpaired two-tailed Student's $t$-tests were performed and $p$ value is indicated on the graph. **f** Depletion of ICMAP1 leads to a complete loss of the ICMT-associated fibers. For panels (**c**)–(**f**), 20 independent tomograms per condition were quantified.

abrogated, when assessed using a red-green invasion assay in which intracellular and extracellular parasite can be distinguished using differential permeabilization and staining[41] (Fig. 4a). Roughly 80% of invasion was observed in the control conditions, but only 40% upon ICMAP2 or ICMAP3[I] depletion following a 30 minutes incubation with host cells. We next assessed microneme secretion via immunodetection of excreted-secreted micronemal antigens (ESA) by western blot[42] (Supplementary Fig. 4d). When compared to the control Tir1 strain, levels of MIC2 adhesin shed in the supernatant fraction were similar in ICMAP2- and ICMAP3[I]-depleted parasites. The absence of a microneme secretion defect is concordant with the normal motility and egress observed. To test if the reduced invasion could be rooted in a defect in rhoptry discharge, we measured this process using the phospho-STAT6 assay (Fig. 4b). Phosphorylation of host cell STAT6 (STAT6-P) and its translocation to the nucleus are a consequence of the injection of the rhoptry kinase, ROP16, into the host cells[43]. Upon depletion of ICMAP2 or ICMAP3[I], the percentage of STAT6-P-positive cells dropped roughly 4-fold but was not completely abrogated. As controls, ASP3 inducible knock-down strain (ASP3-iKD) was previously reported to be fully defective in rhoptry discharge, and its parental strain ΔKu80 was used[44]. To test if the partial defect observed in the inducible knock-down strains could be due to an incomplete downregulation of the proteins, we repeated the experiment with the ΔICMAP2 strain. Here the rhoptry discharge defect was evident but also not complete suggesting that the residual discharge is

not due to an incomplete depletion of the protein (Supplementary Fig. 4e).

## Depletion of ICMAP2 or ICMAP3[I] leads to a defect in rhoptry positioning inside the conoid

The rhoptry discharge defect observed following the depletion of ICMAP2 or ICMAP3[I] could have two likely causes: disruption of the RSA[23] and/or a general defect in rhoptry positioning[45]. To determine whether the RSA was impacted by the absence of ICMAP2 or ICMAP3[I], we used IFA to detect Nd6, an RSA component and indirect marker of the AV[23]. The apical signal of Nd6 was not affected upon ICMAP2 or ICMAP3[I] depletion by IFA (Supplementary Fig. 4f) suggesting intact RSA architecture. This experiment was repeated at the highest resolution in the ICMAP2-mAiD-HA strain using U-ExM and Nd6, visible directly above the ICMTs at an unprecedented resolution, was unaffected in the absence of ICMAP2 (Fig. 4c).

To further investigate the cause of the rhoptry exocytosis defect in the absence of ICMAP2 or ICMAP3[I], we assessed the positioning of the rhoptries by IFA using antibodies against the Armadillo-Repeat Only protein (ARO), which localizes to the surface of the organelle[45] (Supplementary Fig. 4g). In control conditions, ARO staining is observed near the edge of the apical cap stained with anti-ISP1 (Inner membrane complex Sub-compartment Protein 1) antibodies[46]. At this resolution, the bundle of rhoptries was detectable apically with no discernable defect upon ICMAP2 or ICMAP3[I] depletion. Unfortunately,

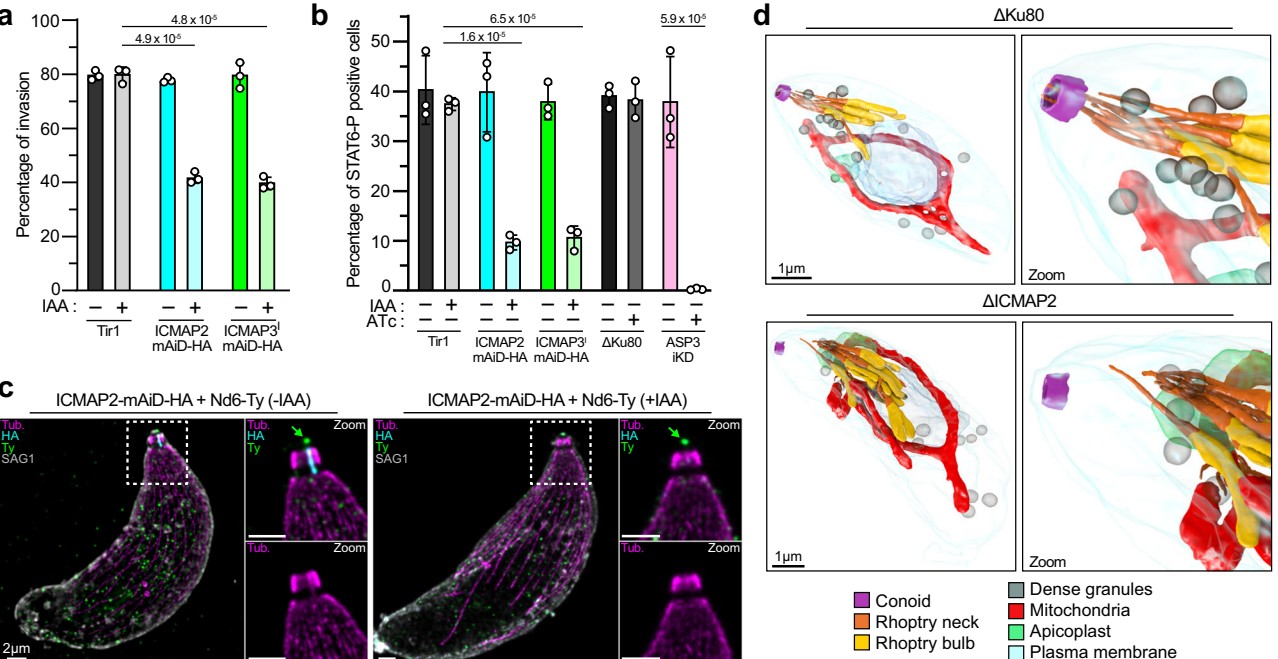

**Fig. 4 | ICMAP2- and ICMAP3$^I$-depletions induce defects in rhoptry discharge and invasion. a** ICMAP2- and ICMAP3$^I$-depleted parasites display a decreased invasion rate via red-green invasion assay. **b** Rhoptry secretion is reduced upon conditional depletion of ICMAP2 and of ICMAP3$^I$. Asp3 is used as a control completely blocked in rhoptry secretion in the presence of ATc with his parental line ΔKu80. For panels (**a**) and (**b**), $n = 3$ independent biological replicates, mean ± SD with individual replicates is presented, and unpaired two-tailed Student's *t*-tests were performed where ns if $P > 5 \times 10^{-2}$. **c** Nd6 at the rhoptry secretion apparatus (AV) is not affected upon ICMAP2 depletion as seen by U-ExM of extracellular parasites. Tub, tubulin. **d** FIB-SEM reconstruction of tachyzoite shows the undocking of the most apical rhoptries in ICMAP2-depleted parasites.

U-ExM was shown to not be suited for fine rhoptry positioning assessment as the apical part of the rhoptry neck cannot be detected using this technique[47]. To overcome this and to increase resolution, we used Focused Ion Beam Scanning Electron Microscopy (FIB-SEM) to assess the fine positioning of the rhoptries. This experiment was performed only in the ΔICMAP2 strain as depletion of ICMAP2 also induces the loss of ICMAP3$^I$. As previously reported, the bundle of rhoptry is apically located and two rhoptry necks are inserted inside the conoid in wild-type parasites[48] (Fig. 4d and Supplementary Movie 3). Upon depletion of ICMAP2, the bundle of organelles was still apically positioned; however, rhoptries were no longer seen to funnel inside the conoid (Fig. 4d and Supplementary Movie 4). Taken together, the role of ICMAP2 and ICMAP3$^I$ in invasion is associated with rhoptry discharge and tentatively linked to the apical positioning of rhoptries and the docking of two rhoptries at the AV.

**Loss of ICMAP2 leads to a disorganization of the ICMTs and their associated structures**

In ICMAP2-depleted parasites, the ICMTs were rarely detectable by U-ExM with anti-tubulin antibodies (Figs. 4c and 5a). Quantification revealed that the ICMTs were visible in 90% of the wild-type parasites observed but only in 10% of those that lost ICMAP2 (Fig. 5a). To rule out potential artifact of antibody accessibility and to exclude that an incomplete depletion of ICMAP2 could lead to spared ICMTs, ΔICMAP2 parasites were examined by negative-stain electron microscopy and again, a substantial loss of ICMTs was recorded (Fig. 5b). Next, ICMAP2-mAiD-HA parasites were scrutinized by cryo-ET which revealed a range of morphological alterations (Fig. 5c and Supplementary Movie 5). The AV was present in 95% of the ICMAP2-depleted parasites (Fig. 5c, d) as previously suggested by the Nd6 staining, but only a single or no rhoptry was docked to it (Fig. 5c–e), confirming the FIB-SEM analysis (Fig. 4d). Moreover, both ICMTs were visible in 60% of the parasites, while another 20% had a single visible ICMT, and no detectable ICMTs in the remaining

20% (Fig. 5c–f). Even when both ICMTs were seen inside the conoid, they were detached from each other and deviated away from the longitudinal axis of the conoid (Fig. 5c). Furthermore, the ICMTs were markedly shorter in the absence of ICMAP2 with a length of around 245 nm (±60 nm) compared to the 376 nm (±59 nm) in control parasites (Fig. 5g). Finally and importantly, in all cases, the MVs were dispersed inside the conoid or into the cytosol in parasites depleted of ICMAP2 (Fig. 5c). Interestingly, in one of the tomograms, we observed an MV attached to the plasma membrane next to the AV, and both vesicles showed an 8-fold rosette on the parasite surface, suggestive of RSA formation on both (Fig. 5h).

**Depletion of ICMAP3$^I$ disrupts rhoptry docking to the AV**

Imaging of ICMAP3$^I$-depleted parasites by U-ExM (Fig. 6a) and negative-stain electron microscopy (Fig. 6b) showed normal, well-positioned ICMTs. Next, cryo-ET on both ICMAP3$^I$ and ICMAP3$^{II}$ mutants confirmed the presence of ICMTs and AV (Fig. 6c, d and Supplementary Movies 6 and 7). Again, the length of ICMTs was reduced upon both ICMAP3$^{II}$ (294 ± 27 nm) and ICMAP3$^I$ (330 ± 39 nm) depletion compared to the control strain (376 ± 59 nm) (Fig. 6e). The number of MVs along the ICMTs was also reduced, possibly as a consequence of ICMTs shortening (Fig. 6f)—while four or five MVs are typically found in controls, ICMAP3$^I$- and ICMAP3$^{II}$-depleted parasites displayed predominantly three or four MVs. The distances between each pair of neighboring MVs (MV-MV distance) and the distance between each MV and the ICMT$_2$ (ICMT-MV distance) remained similar, although an increase in variability was noticed (Supplementary Fig. 5a, b). However, the major difference between the two isoforms of ICMAP3 resided in the fact that ICMAP3$^I$-depleted parasites showed an additional defect in rhoptry docking to the AV (Fig. 6g), explaining the impairment in rhoptry discharge. Intriguingly, upon ICMAP3$^I$ depletion, rhoptry tips could be seen in close contact with aligned MVs, suggesting that in the absence of ICMAP3$^I$, MVs and rhoptries can aberrantly interact (Fig. 6c, left panels and Supplementary Movie 6)

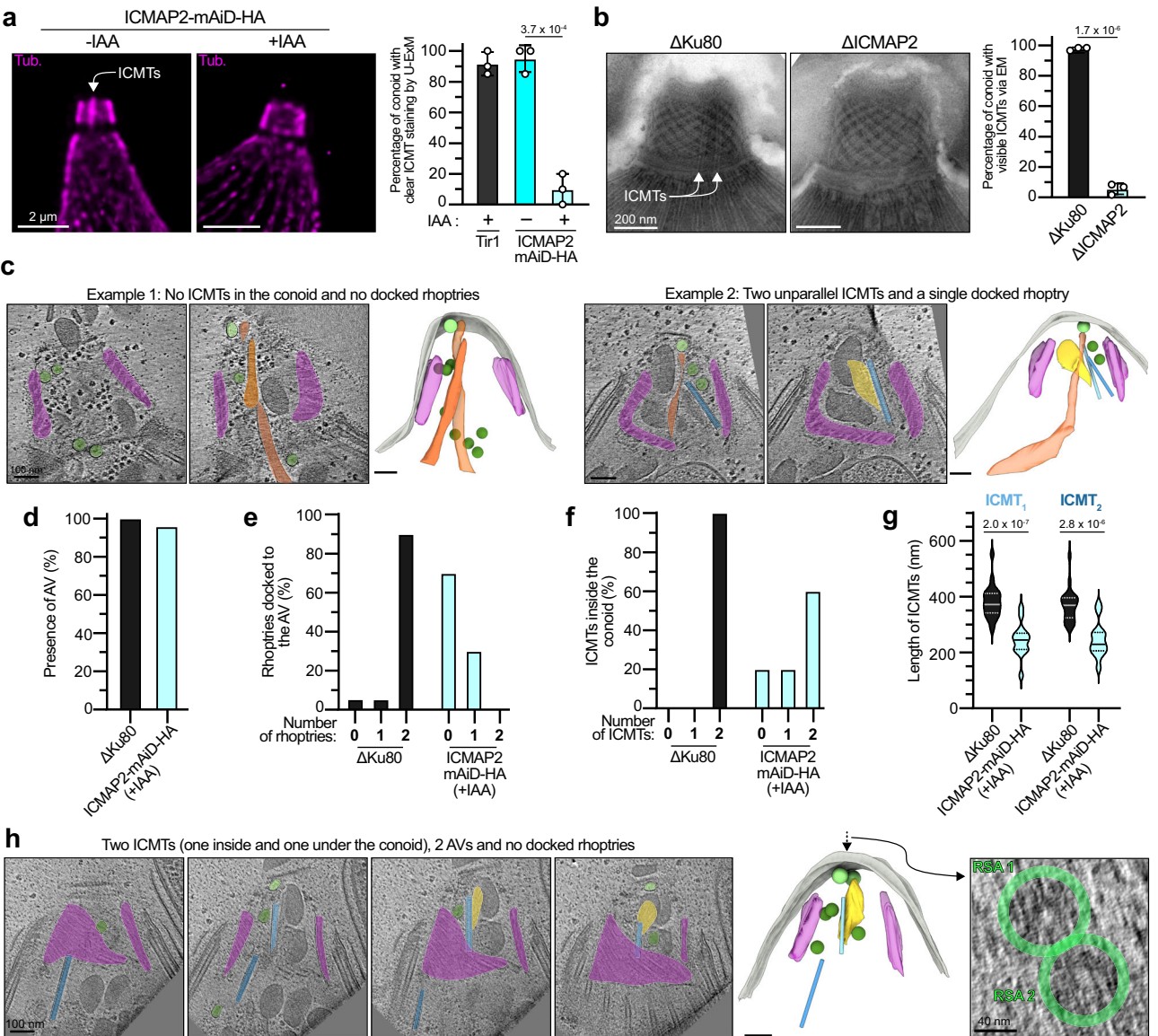

**Fig. 5 | ICMTs, rhoptries and MVs are affected by ICMAP2 depletion. a** ICMTs signal is partially lost upon ICMAP2 depletion in extracellular parasites when assessed by U-ExM. On the left are representative pictures, with a white arrow pointing at the ICMTs staining. Tub tubulin. Mean ± SD and individual replicates are presented (n = 20 for each replicate). Unpaired two-tailed Student's t-tests were performed and p value is indicated on the graph. **b** ICMTs seem lost upon icmap2 knock-out when assessed by EM of extracellular parasites. On the left are representative pictures, with a white arrow pointing at the ICMTs. Mean ± SD and individual replicates are presented (n = 59 for each replicate). Unpaired two-tailed Student's t-tests were performed and the p value is indicated on the graph. **c** Cryo-ET reveals that ICMAP2-depleted parasites show abnormal positioning of MVs, rhoptries and ICMTs. Two typical tomograms are shown. Tomograms at two relevant cross-sections are shown on the left and the associated 3D segmentation is shown on the right. **d** Depletion of ICMAP2 does not affect the presence of the AV. **e** Depletion of ICMAP2 affects the number of rhoptries docked to the AV. **f** Depletion of ICMAP2 affects the number of ICMT observed inside the conoid. **g** Significantly shorter ICMTs are observed in ICMAP2-depleted parasites. For ICMAP2, n = 13 for each ICMT. Unpaired two-tailed Student's t-tests were performed where ns if P > 5 × 10⁻². For panels (**d**)–(**g**), 20 independent tomograms per condition were quantified. For panels (**d**), (**e**) and (**g**), the ΔKu80 control data are the same as the ones presented in Fig. 3. **h** Cryo-ET reveals, in one tomogram of ICMAP2-depleted parasite, the presence of two AVs docked at the plasma membrane. Tomograms at four relevant cross-sections are shown on the left and the associated 3D segmentation is shown next to it. A top view is shown on the right panel, highlighting the presence of two 8-fold rosettes.

## ICMTs are an attribute of coccidian parasites to discharge rhoptries in burst

In ICMAP2-depleted parasites, the overall defect in the architecture of the rhoptry secretion system leads to a significant impairment in invasion (Fig. 7a). *Toxoplasma* belongs to the apicomplexan subgroup of Coccidia that possesses numerous rhoptries presumably allowing several rounds of organelle discharge during invasion and into cells that are not ultimately infected[24]. Upon deletion of ICMAP2, the ICMTs are severely disorganized, offering a unique opportunity to test this hypothesis. To quantify iterative rhoptry discharge prior to invasion,

we detected injected host cells measuring the presence of nuclear STAT6-P, and invaded host cells using GRA3, a marker of the parasitophorous vacuole in intracellular parasites. As previously reported, most STAT6-P-positive HFF cells were invaded and only 5–10% were injected but uninfected (Fig. 7b, c)[24]. As controls, we used ASP3-depleted parasites that are completely blocked in rhoptry discharge[44] and GAC-depleted parasites that are completely blocked in invasion but capable of rhoptry discharge[49]. As expected, neither STAT6-P positive cells nor intracellular parasites were detected when using ASP3-depleted parasites (Fig. 7b, c), while in GAC-depleted parasites

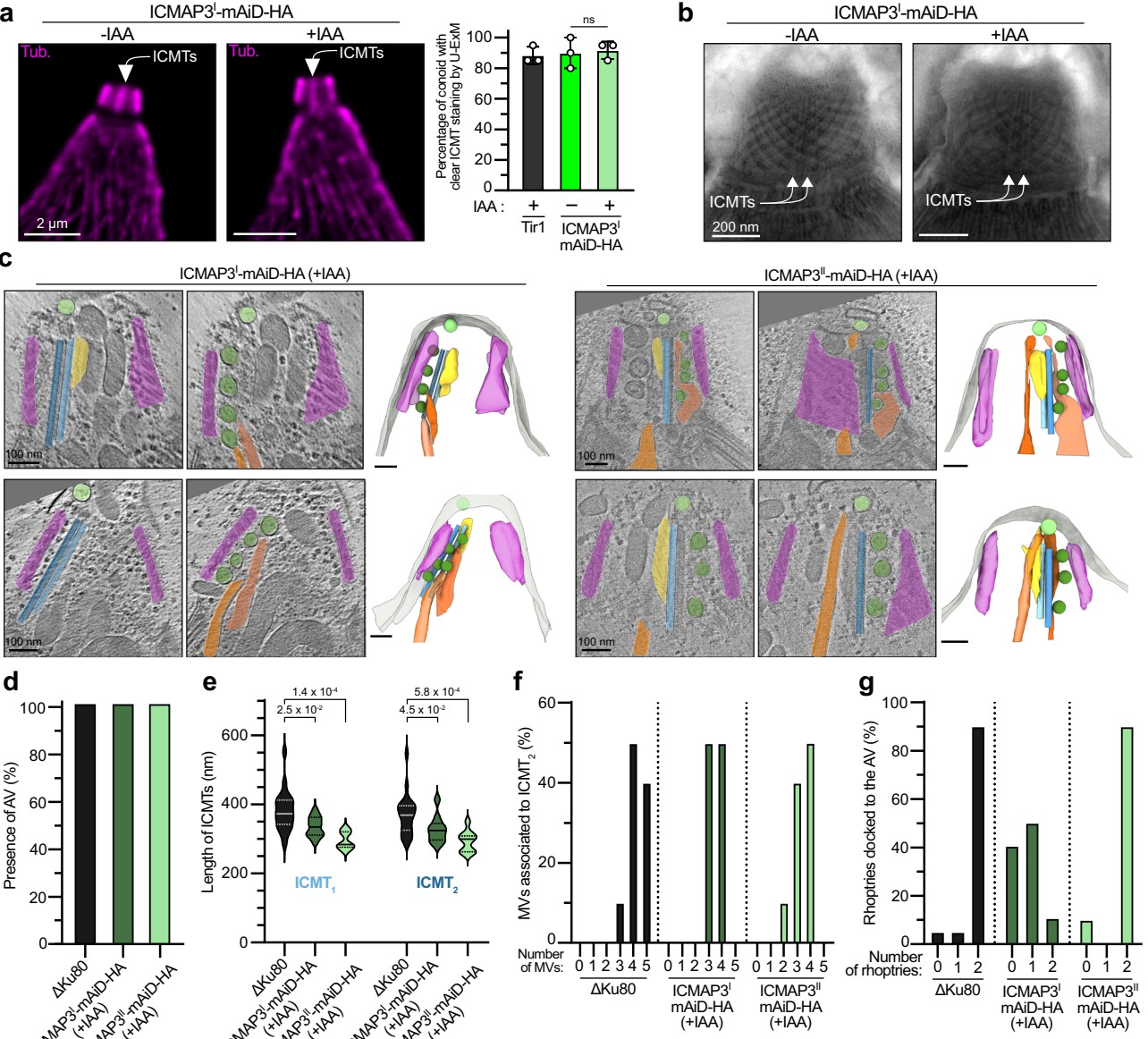

**Fig. 6 | Rhoptry docking is affected by ICMAP3I depletion. a** ICMTs signal is still detected after ICMAP3I depletion when assessed by U-ExM in extracellular parasites. On the left are representative pictures, with a white arrow pointing at the ICMT staining. Tub, tubulin. Mean ± SD and individual replicates are presented ($n = 20$ for each replicate). Unpaired two-tailed Student's $t$-tests were performed where ns if $P > 5 \times 10^{-2}$. **b** ICMTs are present after ICMAP3I depletion when assessed by EM of extracellular parasites. Representative pictures of 3 independent experiments are presented. White arrows point at the ICMTs. **c** Cryo-ET reveals that ICMAP3I-depleted parasites show normal positioning of MVs and ICMTs but a defect of rhoptry docking to the AV while the overall structure appears normal in ICMAP3II-depleted parasites. Tomograms at two relevant cross-sections are shown on the left and the associated 3D segmentation is shown on the right. **d** Depletion of ICMAP3I or ICMAP3II does not affect the presence of the AV. **e** Significantly shorter ICMTs are observed in ICMAP3I- and ICMAP3II-depleted parasites. Unpaired two-tailed Student's $t$-tests were performed, and $p$ values are indicated on the graph. The median of the ICMT size distribution is presented as a solid line, while the first and third quartiles are presented as dotted lines. **f** Depletion of ICMAP3I or ICMAP3II reduces the number of MVs associated with ICMT$_2$. MVs were considered associated with ICMT$_2$ if the distance between the two was shorter than 20 nm. **g** Depletion of ICMAP3I affects the number of rhoptries docked to the AV. For panels (**d**)–(**g**), 10 independent tomograms per condition were quantified. For panels (**d**), (**e**) and (**g**), the ΔKu80 control data are the same as the ones presented in Figs. 3 and 5.

the STAT6-P positive cells were almost exclusively injected but not infected (Fig. 7b, c). Upon ICMAP2 depletion, parasites were concomitantly impaired in invasion and injection with both events roughly halved (Fig. 7c). Then, we tested the impact on virulence in mice of this reduced bystander effect of rhoptry contents on neighboring host cells. It appeared that ICMAP2-depleted parasites were completely avirulent and were unable to elicit seroconversion despite their dispensability in in vitro culture with a residual capacity to invade host cells (Fig. 7d and Supplementary Fig. 6a). Those mice were challenged with ICMAP2-depleted parasites at high titer (10,000 parasites per mice), and again, survived the infection, without raising any antibody response against the parasites (Fig. 7d and Supplementary Fig. 6b). Finally, we assessed if ICMAP2-depleted parasites were capable of forming bradyzoite containing cysts despite their inability to mount a protective immune response. For that, we generated a ΔICMAP2 strain in ME49 background to allow in principle and under appropriate

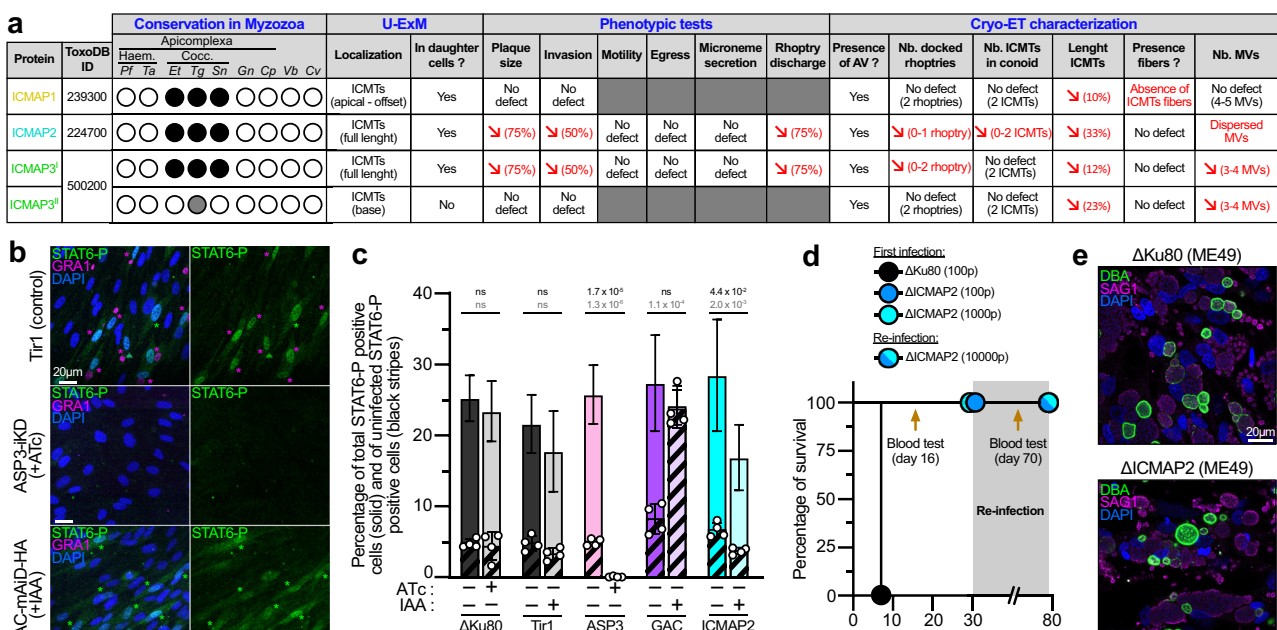

**a**

| Protein | ToxoDB ID | Conservation in Myzozoa | | | | | | | | | | | U-ExM | | Phenotypic tests | | | | | | Cryo-ET characterization | | | | | |
|---|---|---|---|---|---|---|---|---|---|---|---|---|---|---|---|---|---|---|---|---|---|---|---|---|---|---|
| | | Haem. | | Apicomplexa | | | | | | | | | Localization | In daughter cells ? | Plaque size | Invasion | Motility | Egress | Microneme secretion | Rhoptry discharge | Presence of AV ? | Nb. docked rhoptries | Nb. ICMTs in conoid | Lenght ICMTs | Presence fibers ? | Nb. MVs |
| | | Pf | Ta | Cocc. Et | Tg | Sn | Gn | Cp | Vb | Cv | | | | | | | | | | | | | | | | |
| ICMAP1 | 239300 | ○ | ○ | ● | ● | ● | ○ | ○ | ○ | ○ | | | ICMTs (apical - offset) | Yes | No defect | No defect | | | | | Yes | No defect (2 rhoptries) | No defect (2 ICMTs) | ↘ (10%) | Absence of ICMTs fibers | No defect (4-5 MVs) |
| ICMAP2 | 224700 | ○ | ○ | ● | ● | ● | ○ | ○ | ○ | ○ | | | ICMTs (full lenght) | Yes | ↘ (75%) | ↘ (50%) | No defect | No defect | No defect | ↘ (75%) | Yes | ↘ (0-1 rhoptry) | ↘ (0-2 ICMTs) | ↘ (33%) | No defect | Dispersed MVs |
| ICMAP3^I | 500200 | ○ | ○ | ● | ● | ● | ○ | ○ | ○ | ○ | | | ICMTs (full lenght) | Yes | ↘ (75%) | ↘ (50%) | No defect | No defect | No defect | ↘ (75%) | Yes | ↘ (0-2 rhoptry) | No defect (2 ICMTs) | ↘ (12%) | No defect | ↘ (3-4 MVs) |
| ICMAP3^II | | ○ | ○ | ○ | ● | ○ | ○ | ○ | ○ | ○ | | | ICMTs (base) | No | No defect | No defect | | | | | Yes | No defect (2 rhoptries) | No defect (2 ICMTs) | ↘ (23%) | No defect | ↘ (3-4 MVs) |

**Fig. 7 | ICMTs are involved in iterative rhoptry discharge. a** Table recapitulating the conservation and phenotypes observed for the ICMAP proteins. Cocc., Coccidia; Haem., Heamosporidia; Pf, *Plasmodium falciparum*; Ta, *Theileria annulata*; Et, *Eimeria tenella*; Tg, *Toxoplasma gondii*; Sn, *Sarcocystis neurona*; Gn, *Gregarina niphandrodes*; Cp, *Cryptosporidium parvum*; Vb, *Vitrella brassicaformis*; Cv, *Chromera velia*. For conservation, ortholog presence (black circle) or absence (open circle) in selected myzozoan taxa is shown (gray circle when the flawed genome annotation does not allow proper conservation assessment). In red are highlighted the main phenotypes. **b** Representative pictures of the "injected-invaded" assay for three control strains. Magenta asterisks indicate infected STAT6-P positive cells. Green asterisks indicate uninfected STAT6-P positive cells (injected-only cells). **c** Quantifications of the "injected-invaded" assay. Unpaired two-tailed Student's *t*-

tests were performed where ns if $P > 5 \times 10^{-2}$ ($n = 4$ independent biological replicates). For total rhoptry discharge (solid bar) only the mean ± SD is represented. For the "injected-only" (black stripes) fraction, mean ± SD with individual replicates is presented. Top *p* value corresponds to total rhoptry discharge and bottom *p* value corresponds to "injected-only". **d** Mice survival and subsequent re-infection. For each strain, the number of parasites injected is presented in parentheses. For the first infection, 5 mice per condition were injected intraperitoneally. For the challenge infection, 4 mice were infected. Blood tests were carried out to check for seroconversion. **e** Myotube infection demonstrates the ability of the different ME49 strains to convert into bradyzoites and form cysts (DBA - *Dolichos biflorus* agglutinin lectin: a marker of cyst wall). Representative pictures of 3 independent experiments are presented.

culture conditions[50], tissue cysts formation (Supplementary Fig. 6c). Like its counterpart in an RH background, the ΔICMAP2 (ME49) strain displayed a clear decrease in fitness by plaque assay (Supplementary Fig. 6d) as well as a loss of virulence and absence of seroconversion in the mouse model of infection (Supplementary Fig. 6e). To assess tissue cysts formation, myotubes were infected with parental ΔKu80 (ME49) or ΔICMAP2 (ME49) strains and cultured in absence of $CO_2$. Interestingly, the ΔICMAP2 (ME49) parasites were perfectly able to convert into bradyzoites, expressing bradyzoites-specific markers, p36 and SAG4 as well as the late marker p21 (Supplementary Fig. 6f). The formation of large cysts was confirmed using anti-DBA (cyst wall marker) lectin (Fig. 7e and Supplementary Fig. 6f).

## Discussion

The ICMTs are at the center of the apical complex in coccidian parasites and the data presented here establish their role in ensuring multiple rounds of rhoptry discharge. ICMAP1 was the only protein previously described exclusively at the ICMTs of *T. gondii*[14]. At the time, the immunogold labeling on deoxycholate-extracted parasites was the only method able to assign proteins to the ICMTs. U-ExM allowed us to demonstrate its localization adjacent to the ICMTs suggesting a link with the $ICMT_1$-associated fibrous material. Cryo-ET on ICMAP1-depleted parasites was then able to confirm this relationship. ICMAP1 has been shown to bind to microtubules, when expressed in mammalian cells, and to decorate microtubules polymerized in vitro[14]. Collectively, the results suggest that the fibrous material associated with the ICMTs is composed of tubulin polymers. The lack of staining with antibodies against acetylated tubulin or α- and β-tubulin by U-ExM

could be due to an atypical form of α- and β-tubulin polymers not recognized by those antibodies or steric hindrance due to modification or presence of additional microtubule-binding proteins. The origin and role of these fibers remain to be elucidated. In a recent cryo-ET study, a third ICMT has been described[29]. While this elusive third ICMT has never been observed in our tomograms or any prior EM-based studies[4,22], a link between fibrous material and third ICMT cannot be excluded. Our study used two complementary techniques, FIB-SEM and cryo-ET, to examine the role of the ICMAPs. FIB-SEM is performed on fixed samples at a comparatively low resolution but allows a greater membrane contrast of large structures. Cryo-ET on the other hand is performed on native samples preserving cellular structures and at higher resolution allowing the analysis of smaller structures.

Recently, Gui et al. determined the polarity of the ICMTs using the chirality of their cross-sections observed by cryo-ET[29,51]. They concluded that the distal part of the ICMTs, where we show ICMAP3^II specifically localizes, corresponds to the plus-end (fast-growing end) of the microtubules. With its distal localization, it could be hypothesized that the distal part of the ICMTs might be unstable and depolymerized in the absence of ICMAP3^II, resulting in a shortening of the ICMTs. ICMAP3^II could therefore be an important player in the ICMTs dynamics. ICMAP3^II amino acids sequence annotation is hampered by two large stretches and repeated sequences, yet identification of interacting partners involved at the ICMTs growing tip should help unraveling the unique biology of these unusually short and stable ICMTs compared to canonical microtubules. The shortening of the ICMTs was concomitant with a lower number of associated MVs, suggesting that the length of the ICMTs might be an essential factor for

the number of associated MVs. Of relevance, when the apical tip of *Eimeria tenella* sporozoites has been described using electron tomography, a disparity in the length of $ICMT_1$ and $ICMT_2$ was striking. While $ICMT_1$ seems to span roughly one conoid-length, $ICMT_2$ associated with MVs was several times longer, extending inside the parasite cytosol[35]. We did not identify an $ICMAP3^{II}$ homolog in *Eimeria*; however, its genome contains a large number of repetitive sequences, which has hindered annotation and may obscure the presence of some genes[52].

While ICMAP1 and $ICMAP3^{II}$ phenotypes could be pinpointed to a loss of fibrous material and shortening of the ICMTs, respectively, the phenotype induced by the depletion of ICMAP2 and $ICMAP3^{I}$ were more heterogeneous and severe, affecting the numbers of rhoptries docked to the AV, the length of the ICMTs, as well as their positioning in the case ICMAP2. Of note, it is likely that the shortening of the ICMTs observed after ICMAP2 depletion is a consequence of the concomitant loss of $ICMAP1/ICMAP3^{I}/ICMAP3^{II}$ in those parasites. The positioning of the ICMTs seemingly untethered at the center of the conoid is still enigmatic. In original EM pictures showing cross-sections of the conoid, the ICMTs are seen quite a distance away from the conoid walls[4]. The main hypothesis to explain this observation has been that the ICMTs could be anchored at the PCRs or to the conoid wall through an arm-like structure[4]. While the PCR hypothesis is appealing, recent studies show that mutants lacking the PCRs display a normal ICMTs positioning inside the conoid[12]. In parallel, we never observed any arm-like structure or connections linking the ICMTs and the conoid in the tomograms, leaving this problematic unanswered. In addition to their positioning inside the conoid, the linking of the two ICMTs together was specifically disrupted after ICMAP2 depletion, putting the protein as a prime candidate for this function. Ultimately, the sum of those phenotypes hindered rhoptry anchoring and discharge, making the disentanglement of the causes and consequences difficult. Importantly, the rhoptry discharge defect was not complete and could be explained by the fact that ICMAP2- or $ICMAP3^{I}$-depleted parasites still display AV-anchored rhoptry. In addition, residual invasion in those mutants suggests that this process could be carried out successfully even if a single rhoptry is associated with the AV.

ICMTs are only conserved in organisms possessing more than two rhoptries, fueling the hypothesis that they ensure several rounds of rhoptry discharge. Of considerable relevance, *T. gondii* uses rhoptry contents to subvert host cellular functions and do it not only during invasion but also in the neighboring non-infected cells[24]. The secreted rhoptry proteins such as ROP16 are not only bystanders but critical players modulating the host immune response[53]. The ICMTs therefore likely provide the scaffold for iterative discharge of the rhoptries during infection, which presumably contributes to the formation of protective niche to achieve chronic infection and thus ensure transmission[54]. Apicomplexans such as *Plasmodium* and *Cryptosporidium* might not need to subvert neighboring host cells through iterative rhoptry discharge. While *Plasmodium* circulates continuously in the blood stream and only modifies the infected cells to limit splenic clearance, *Cryptosporidium* infection leads to rapid destruction of the intestinal epithelium and shedding in the environment. Contrastingly, *Toxoplasma* infect deep tissues to establish a chronic infection and requires the modulation of the environment and the immune response to avoid killing the host[55]. This in turn could explain why iterative rhoptry discharge is only found in *Toxoplasma* and parasites with similar lifestyle. The assessment of rhoptry discharge with and without invasion revealed that ICMAP2 contributes to both events, even if a stronger reduction in discharge without invasion could be expected after such dramatic phenotype highlighted by cryo-ET. More sensitive assay will have to be developed to assess precisely the injection without infection events and importantly under physiological conditions in vivo. As reported by Koshy et al., the uninfected-injected cells are more frequent in the mouse brain than in tissue culture of HFFs[24]

stressing the important role of rhoptries in the manipulation of host environment in vivo. Mouse infection conducted here revealed that parasites mutants lacking ICMAP2 are still able to invade tissue culture but are completely avirulent in vivo. Remarkably, this mutant failed to trigger seroconversion in vivo, suggesting a fast clearance of those parasites by the innate immune system. Lack of seroconversion was previously reported for two mutants lacking ARO or SORTLR both of which show complete loss of rhoptry discharge[55,56]. ARO or SORTLR were reported to interfere with T-helper 1–dependent adaptive immunity and hampers the function of natural killer T-cell–mediated innate immunity[55]. Conversely, a lack of seroconversion was never observed in mutants able to secrete their rhoptries but with an invasion defect[57]. In this context, ICMAP2-depleted mutant provides compelling evidence that the capacity of iterative rhoptry discharge, but not invasion, is responsible for seroconversion and protective immunity. Of relevance, a recent study showed that *T. gondii* has the capacity to manipulate, in a strain-specific and ROP16-dependant manner, the host to generate a favorable encystment environment[58]. Also, the cyst-forming ME49 strain lacking ICMAP2 is avirulent in mouse infection and also fails to induce sterile immunity, although the mutant is able to form cysts in myotubes in vitro. At this stage, the results support the role of the ICMTs as an essential virulence factor, enabling iterative rhoptry discharge to control the host immune response during infection and further work is needed to address the role in the establishment of a suitable niche for chronic infection.

There are many remaining questions to understand the whole complexity of the machinery orchestrating rhoptry discharge in coccidians. We still do not know the origin of the MVs and AV, when they are formed, and how MVs are docked and transported along the ICMTs. In this context, it was striking to observe the docking of a MV to the parasite plasma membrane and forming a new RSA (Fig. 6h). This event was only seen in the absence of ICMAP2 where the MVs are no longer organized along the ICMT, suggesting that ICMTs arrange the MVs to prevent their premature attachment to the plasma membrane at the tip of the parasite. Similarly, the presence of rhoptry docked to aligned MVs in the absence of $ICMAP3^{I}$ was unique to this mutant (Fig. 5c). These observations taken together, in addition to the similar sizes and appearances of MVs and AVs[20,27], suggest that MVs are most likely future AV that will be used for subsequent rounds of secretion.

Overall, the identification and functional characterization of the ICMAPs provided a significant step forward to better understand the biology of the ICMTs. Of relevance, additional proteins have also been localized at least in part to the ICMTs, namely TLAP3 and TrxL1[59,60]. Moreover, new insight on the microtubule inner proteins (MIPs) was recently uncovered by cryo-EM on *T. gondii* SPMTs[28]. The use of cryo-EM on purified ICMTs would be of considerable value to discover the extent of ICMT-specific MIPs. The presence of ICMAP2 at the centrioles of mature daughter cells asks the question of what defines ICMAPs' specificity to the ICMTs compared to other tubulin-based structures. Worth noting, ICMAP1 has also been described to bind the centrioles but only when overexpressed[14]. How these various proteins can target specific tubulin-compartment might depend on several factors such as tight control of the protein abundance, timing of expression, specific post-translational modifications of the tubulin, or specific tubulin isoforms. The only exotic exception of ICMTs presence outside of coccidians, is the description of two microtubules in the lumen of *Chromera velia* pseudoconoid[61]. Those ICMTs were described near membrane-bound organelles, evocative of apicomplexan rhoptries. *Chromera velia* could therefore be the source of captivating evolutionary insights concerning the biology of ICMTs. Understanding high-resolution structural details and evolution of the ICMT and associated proteins are important and yet challenging open questions that will ultimately provide crucial knowledge toward the fight against these important pathogens.

## Methods

### Genes of interest and strategies design

Gene maps were retrieved from the ToxoDB[62] website (www.toxodb. org) using the following accession numbers: ICMAP1 (TGME49_239300), ICMAP2 (TGME49_224700), ICMAP3[I] (TGME49_500200 in ToxoDB version 66 and TGME49_285150 in previous versions), ICMAP3[II] (TGME49_500200 in ToxoDB version 66 and TGME49_285150-285140 in previous versions), Nd6 (TGME49_248640). To target the endogenous locus of the different genes of interest, gRNAs (guide RNA) were designed using the EuPaGDT[63] online tool (www.grna.ctegd.uga.edu).

To generate the ΔICMAP3 strain, the whole TGME49_500200 sequence (formerly spanning both TGME49_285150 and TGME49_285140 models) was deleted and replaced by a resistance cassette. To generate the TetO-myc-ICMAP3 strain, the TetO-myc cassette was inserted at the beginning (5' UTR) of the TGME49_500200 (formerly TGME49_285150) model. The ICMAP3[I] strain was generated by adding the mAiD-HA cassette at the end (3' UTR) of the formerly TGME49_285150 model. The ICMAP3[II] strain was generated by adding the mAiD-HA cassette at the end (3' UTR) of the formerly TGME49_285140 model.

### Parasite culture

*T. gondii* tachyzoites were cultured in HFFs (ATCC) with Dulbecco's Modified Eagle's Medium (DMEM, Gibco) supplemented with 5% of fetal calf serum (FCS, Gibco), 2 mM glutamine and 25 μg/mL gentamicin (Gibco).

The RHΔKu80 (called ΔKu80) strain was used to generate clean knock-out lines. The RHΔKu80ΔTir1 (called Tir1) strain was used to generate conditional knock-down lines using the auxin-induced degron system[37].

Auxin-induced degron fused protein depletion (mAiD strains) was achieved by addition of 500 μM of auxin (IAA)[64] for at least 12 h. Depletion via the Tet-inducible system (iKD strains) was achieved by addition of 1 μg/mL of anhydrotetracycline (ATc)[65].

### DNA constructs and transfections

Specific gRNAs were generated via amplification with the Q5 site-directed mutagenesis kit (New England Biolabs) of the pSAG1::Cas9-U6::sgUPRT template[66]. PCR products for all the -mAiD-HA or -Ty tag fusions containing specific flanking regions were generated from KOD polymerase (Novagen, Merck) using templates and oligos described in Supplementary Data 1. For the ΔICMAP2 line generation, the DHFR resistance cassette was inserted in place of the endogenous locus using two gRNAs targeting the endogenous locus.

Tachyzoites were transfected by electroporation as previously described[67]. For each transfection, 40 μg of gRNA and 100 μL (2 × 50 μL reaction) of KOD polymerase PCR products were used. Parasites carrying an HXGPRT cassette were selected with 25 mg/mL of mycophenolic acid and 50 mg/mL of xanthine. Parasites carrying a DHFR cassette were selected with 1 μg/mL of pyrimethamine.

### Immunofluorescence assay (IFA)

HFF seeded on coverslips were infected with *T. gondii* tachyzoites and grown at 37 °C for 15–24 h. Parasites were then fixed with 4% PFA/ 0.05% glutaraldehyde (PFA-GA) in PBS or ice-cold methanol, neutralized in 0.1 M glycine-PBS for 3–5 min in the case of PFA-GA fixation and blocked in PBS/5% BSA. Antibodies were incubated for 1 h in PBS/ 2% BSA and washes in PBS were conducted between primary and secondary antibody incubation. Coverslips were mounted in DAPI Fluoromount-G (Southern Biotech).

Images were obtained with a Zeiss laser scanning confocal microscope (LSM700 using objective apochromat 63×/1.4 oil). Images were then processed for publication with the ImageJ software. All fixation methods, antibodies dilutions and providers are recapitulated in Supplementary Data 1.

### Ultrastructure expansion microscopy (U-ExM)

U-ExM was performed using published detailed protocols for *T. gondii*[38] with few modifications and optimizations measures detailed here. For intracellular conditions, non-confluent HFF seeded on coverslips were infected with tachyzoite and grown for 18–20 h. For extracellular conditions, freshly egressed tachyzoites were resuspended in warm PBS containing BIPPO to stimulate conoid extrusion[12] and sedimented in Poly-D-lysine (Gibco) coated coverslips. By default, the parasites were not fixed prior to the 0.7% formaldehyde/1% acrylamide (FA/AA) bath (3 h at 37 °C). Only ICMAP3[II] required an ice-cold methanol fixation (7 min at −20 °C) before the FA/AA bath to be detected. The coverslips were then used to cover drops of monomer solution (19% sodium acrylate/10% acrylamide/0.1% bis-acrylamide) supplemented with 0.5% ammonium persulfate (APS) and 0.5% tetramethylethylenediamine (TEMED) to initiate the gel polymerization. After an incubation of 30 min at 37 °C, the fully polymerized gels were detached from the coverslip and incubated for 1 h 30 min at 95 °C in denaturation buffer (200 mM SDS, 200 mM NaCl, 50 mM Tris in water, pH 9). Gels were then expanded overnight in successive baths of water. The next day the gels were shrunk in PBS, cut, and incubated for 3 h at 37 °C with agitation in a mix of appropriate primary antibody diluted in freshly prepared PBS-BSA 2%. After three washes of 10 min in PBS-Tween 0.1%, the gels were incubated for 2 h at 37 °C (with agitation and in the dark) in a mix of appropriate secondary antibodies. After three washes of 10 min in PBS-Tween 0.1%, the gels were expanded overnight in water. Imaging was performed using a Leica TCS SP8 inverted microscope. The microscope was used with an HC PL Apo 100×/1.40 Oil CS2 objective and with HyD detectors. Z-stack was acquired with the Leica LAS X software and deconvolved with the built-in "Lightning" mode. Images were then processed with the ImageJ software and maximum projections were used for publication.

The plotting of the intensity profile was generated using Leica LAS X software and its built-in Quantify and Line profile function. Numerical data were then exported in GraphPad to generate the graphical representations.

### Plaque assay and quantification

HFF monolayer were infected with freshly egressed parasites and grown for 7 days at 37 °C. Cells were then fixed with Paraformaldehyde 4% – glutaraldehyde 0.005% (PFA-GA), subsequently neutralized in 0.1 M glycine/PBS. The cells were then washed in PBS and stained with crystal violet. For measurements of the plaque area, images of the monolayers were analyzed using the ImageJ software.

### Fractionation assay

Freshly egressed tachyzoites were harvested, washed in PBS, and then resuspended in either PBS, PBS/1% Triton-X-100, PBS/0.1 M $Na_2CO_3$ at pH 11.5 or PBS/1% SDS. Parasites were lysed by 4–5 cycles of freeze and thaw in liquid nitrogen and incubated on ice for 30 min. Pellets and supernatants were then separated by centrifugation at 4 °C for 30 min at 15,000 × g. Samples were then run by western blot and representative pictures are displayed in the manuscript. PBS/1% SDS is a control that should solubilize all proteins. PBS/0.1 M $Na_2CO_3$ should solubilize protein in tight complexes. PBS/1% Triton-X-100 should solubilize membrane-bound proteins such as GAP45. Cytosolic proteins, such as catalase, should be solubilized in PBS.

### Cryo-electron tomography

*T. gondii* tachyzoites were prepared for cryo-ET as previously described[20]. Briefly, 4 μL containing around 4 million tachyzoites mixed with 10 nm gold fiducials were loaded onto an EM grid for plunge freezing into a liquid ethane/propane mixture on an EM GP2 automatic plunger (Leica Microsystems, Wetzlar, Germany) after 4 s of back blotting. Images were collected on a Thermo Fisher Krios G3i 300 keV field emission cryo-transmission electron microscope equipped with a Volta

phase plate[68] and an energy filter. Dose-fractionated imaging was performed using the SerialEM software on a K3 direct electron detector[69] (Gatan) operated in electron-counted mode. Tilt-series were collected with a span of 120° (−60° to +60°; bi-directional scheme) with 2° increments at a magnification of 33,000×, a nominal pixel size of 2.65 Å and a defocus range of −1 to −4 μm. Tilt-series were aligned using the gold fiducials and reconstructed into tomograms using the IMOD software[70]. Motion correction of images was done using the Alignframe function in IMOD. Segmentation and videos were generated in IMOD, and UCSF ChimeraX[71] was used for display purposes. Note that when the parasite plasma membrane was lysed due to the sample preparation, the estimated plasma membrane location was outlined. All quantifications were performed in IMOD. All tomograms without overlay can be found in Supplementary Fig. 7. In total, 20 tomograms were analyzed for each WT, ICMAP1 and ICMAP2 mutants, while 10 were analyzed for each ICMAP3 mutants. Note that MVs and AVs are represented as simple spherical objects in 3D segmentations to display their distribution, while statistical measurements of MV-ICMT$_2$ and MV-MV distances are conducted on the EM densities in raw tomograms.

### Gliding (trail-deposition) assay

The assay was performed as described previously[72]. Shortly, parasites were resuspended in a calcium saline solution before sedimentation on poly-L-lysine coated coverslips. Parasites were incubated at 37 °C for 15 min to allow gliding. The parasites were subsequently fixed with PFA-GA and stained with anti-SAG1 antibodies, in the absence of Triton X100 to maintain membrane integrity.

### Red-green invasion assay

HFF monolayer were infected with *T. gondii* tachyzoites, sedimented at 1000 × g for 1 min and incubated at 37 °C for 30 min to allow invasion. Cells were then fixed with PFA-GA, blocked in 5% PBS-BSA for 20 min, incubated 1 h with anti-SAG1 antibodies and washed three times in PBS. Cells and antibodies were then fixed with 1% formaldehyde for 7 min and permeabilized with 0.2% Triton X100 in PBS for 20 min. Finally, cells were incubated with anti-GAP45 antibodies, washed three times, and incubated with appropriate secondary antibodies. Two hundred parasites were counted for each condition and over three independent biological replicates ($n = 3$).

### Induced-egress assay

Coverslips with HFF monolayer were infected with *T. gondii* tachyzoites, centrifuged at 1000 × g for 1 min and incubated at 37 °C for 30 h to allow growth. Cells were then incubated with DMEM media containing BIPPO or DMSO for 10 min at 37 °C. Coverslips were then fixed with PFA-GA and labeled as described previously, with anti-GAP45 and anti-GRA3 antibodies. Two hundred vacuoles were counted for each condition and over three independent biological replicates ($n = 3$).

### Microneme secretion

Freshly egressed parasites were washed twice in warm DMEM media. They were then pelleted and resuspended in media containing 2% Ethanol (EtOH) or 2% DMSO and incubated at 37 °C for 10 min. Pellets and supernatant (ESA) were then separated by centrifugation at 2000 × g. The ESA fraction was then centrifuged at 5000 × g to remove cell debris. Pellet fractions were washed once in PBS to remove any ESA remaining. Pellets and ESA were then analyzed by western blot using anti-MIC2, anti-Catalase and anti-GRA1 antibodies. The assay was repeated three times and a representative replicate is presented in the manuscript. Pictures were acquired using the ImageLab software (Bio-Rad).

### Phospho-STAT6 rhoptry secretion assay

Freshly egressed parasites ($5 \times 10^5$) pretreated ±IAA or ±ATc were used to infect HFF monolayer-coated coverslips. After a short centrifugation (1000 × g for 50 s) cells were incubated in a warm bath at 37 °C for 20 min. After cold-methanol fixation (7 min at −20 °C), blocking with 5% Bovine Serum Albumin in PBS was performed and immuno-detection was performed using anti-phospho-STAT6 antibody (Cell signaling 9361; 1/400). The experiments were performed in triplicate and >300 host cells nuclei were counted each time.

**For "injection-infection" experiment.** HFF cell coated coverslips were infected with $5 \times 10^4$ freshly egressed parasites (MOI = 0.2), pretreated 48 h ±IAA or ±ATc. Cells were then incubated at 37 °C for 18 h. After cold methanol fixation and blocking with 5% Bovine Serum Albumin in PBS, immuno-detection was performed using anti-phospho-STAT6 antibody and anti-GRA3 (in house hybridoma; 1/100). The experiments were done in triplicate and for each condition 20 fields of view were quantified.

### Focused ion beam scanning electron microscopy (FIB-SEM)

HFF cells infected with parasites were grown in monolayer on round (12 mm in diameter) glass coverslips and were fixed with 2.5% glutaraldehyde (GA) (Electron Microscopy Sciences) and 2% PAF (paraformaldehyde) (Electron Microscopy Sciences) in 0.1 M sodium cacodylate buffer at pH 7.4 for 1 h at room temperature. Cells were extensively washed with 0.1 M sodium cacodylate buffer, pH 7.4 and post-fixed with 1% osmium tetroxide (Electron Microscopy Sciences) and 1.5% potassium ferrocyanide in 0.1 M sodium cacodylate buffer, pH 7.4 for 1 h, followed by 1% osmium tetroxide (Electron Microscopy Sciences) in 0.1 M sodium cacodylate buffer pH 7.4 alone for 1 h. Cells were then washed in double distilled water twice for 5 min each wash and block stained with aqueous 1% uranyl acetate (Electron Microscopy Sciences) at 4 °C overnight. The next day, samples were washed twice for 5 min in double distilled water and dehydrated in graded ethanol series (2 × 50%, 70%, 90%, 95%, and 2 × absolute ethanol) for 10 min each wash. After dehydration, cells were infiltrated with graded series of Durcupan resin (Electron Microscopy Sciences) diluted with ethanol at 1:2, 1:1, 2:1 for 30 min each, and twice with pure Durcupan for 30 min each. Cells were infiltrated with fresh Durcupan resin for additional 2–4 h. Coverslip with grown cells faced down was placed on 1 mm-high silicone ring (used as spacer) filled with fresh resin, which was placed on a glass slide coated with mold-separating agent. This flat sandwich was then polymerized at 65 °C overnight in the oven. The glass coverslip was removed from the cured resin disk by immersing alternately into hot (60 °C) water and liquid nitrogen until the glass parted.

The selected parasitophorous vacuole was marked on the surface of the resin block by a laser microdissection microscope (Leica Microsystems). Either whole resin block or large cutout area containing the region of interest was glued onto a flat aluminum SEM stub with superglue and silver conductive paste was applied on each side of the resin block to ensure the conductivity within SEM. Finally, the mounted sample was coated with a 20-nm-thick layer of gold.

The samples were imaged inside a FEI Helios NanoLab G3 UC DualBeam microscope (FEI). Ion beam was used in conjunction with a gas injection system to deposit a thick (~1.5 μm) layer of platinum on the top surface of the sample above the region of interest to reduce the FIB milling artifacts. The imaging surface was exposed by creating the front trench using 21 nA of focused ion beam current at 30 kV voltage and subsequently two side trenches were created using the same parameters. AutoSlice and View G3 software (FEI) were used to acquire the serial SEM images. Focused ion beam at a current of 2.5 pA and 30 kV of acceleration voltage was applied to mill 10 nm layer from imaging face and freshly exposed surface was imaged with a backscattered electron beam current of 400 pA at an acceleration voltage of 2 kV, the dwell time of 6 μsec/pixel and at the resolution of 4 nm/pixel.

For the 3D reconstruction, serial images through the selected region of interest were combined into single image stacks and aligned using the FIJI program (fiji.sc/). After alignment, images were scaled

down to have volume with isotropic pixel properties of the 10 nm/pixel in all *x*-, *y*-, and *z*-dimension. Semi-automated approach using Ilastik software (ilastik.org) was used for segmentation and 3D reconstruction. Final 3D models were visualized, and movies rendered using the Blender program (v.2.79; blender.org).

### Negative-stain electron microscopy

As described previously[47], extracellular parasites were pelleted in PBS. Conoid extrusion was induced by incubation with 40 μL of PBS/BIPPO for 5 min at 37 °C. 5 μL of the sample was applied on glow-discharged 200-mesh Cu electron microscopy grid for 5 min. The excess of the sample was removed by blotting with filter paper and immediately washed three times on drops of double distilled water. Finally, the sample was negatively stained with 0.5% aqueous solution of phosphotungstic acid (PTA) for 20 s then blotted with filter paper and air dried. Electron micrographs of parasites apical poles were collected with Tecnai 20 TEM (FEI, Netherland) operated at 80 kV acceleration voltage equipped with side mounted CCD camera (MegaView III, Olympus Imaging Systems) controlled by iTEM software (Olympus Imaging Systems).

For quantification, the preparation of the samples was performed in a blind manner. After acquisition, the names of TEM images were randomized using inhouse written MatLab script and these randomized sets of images were assessed by naive users to assign the presence or absence of the intraconoidal microtubules.

### Animal experimentation

All animal experiments were conducted with the authorization numbers GE3-20 and GE272, according to the guidelines and regulations issued by the Swiss Federal Veterinary Office. The authorizations were issued through the director of the animal experimentation department of the University of Geneva (Daniele Roppolo) and approved by the cantonal veterinary officer (Dr Michel Rérat).

All animals were housed at the University of Geneva in a room with day/night cycle of 12 h/12 h and constant ambient temperature of 22 °C and 35% humidity.

Freshly egressed Δku80 and ΔICMAP2 were counted and diluted in DMEM without supplements. Parasites were injected intraperitoneally in 7-week-old female CD1 mice (Charles River). Five mice per group were divided as follows: 100 parasites of RH parental, 100 and 1000 parasites for ΔICMAP2, for a total of 15 mice. Plaque assays to determine the number of infectious parasites were set up in duplicate. Mice were monitored daily and sacrificed at the onset of signs of acute infection (ruffled fur, difficulty moving, isolation). Mice infected with ΔICMAP2 parasites never showed symptoms of infection and were not seroconverted when tested 16 days post infection. Given the absence of seroconversion in the surviving mice, the challenge infection was not carried out with virulent Δku80 parasites that would have caused certain death and unnecessary suffering. Challenge infection was conducted on 4 surviving mice.

### Myoblast and myotube culture

LHCN-M2 myoblasts were purchased from Evercyte (Cat. No. CkHT-040-231-2) and cultured at 37 °C and 5% $CO_2$ in a humidified incubator. The growth medium is based on the protocol from the manufacturer which consisted of four parts of Dulbecco's Modified Eagle's Medium (DMEM, 4.5 g/L glucose, Gibco) and one part of Medium 199 (M199) (Gibco), supplemented with 15% Fetal Bovine serum (FBS) (Gibco), 20 mM HEPES (PanReac Applichem), 0.03 μg/mL Zinc sulfate (Sigma-Aldrich), 1.4 μg/mL Vitamin B12 (Sigma-Aldrich), 0.55 μg/mL Dexamethasone (Sigma-Aldrich), 2.5 ng/mL HGF (Millipore), 10 ng/mL bFGF (Thermo Fisher Scientific) and 100 U/mL penicillin/streptomycin (Gibco).

To induce myotubes differentiation following the protocol previously described in ref. 50, myoblasts were cultured until they reached a 50–70% confluency, followed by a change to differentiation medium consisting on DMEM (4.5 g/L glucose, Gibco)), supplemented with 2% Horse serum (Brunschwig), 10 μg/mL human insulin (Sigma-Aldrich), 5 μg/mL Holo-transferrin (Sigma-Aldrich), 10 nM sodium selenite (Sigma-Aldrich) and 100 U/mL penicillin/streptomycin. The cells were cultured at 37 °C and 5% $CO_2$ in a humidified incubator for 5–7 days.

### Tissue cyst culture

Differentiation of tachyzoites into tissue cysts was done under $CO_2$ depletion as described in ref. 50. The cyst medium consisted of Roswell Park Memorial Institute 1640 Medium (RPMI, without glucose, Sigma-Aldrich), supplemented with 5 mM glucose, 2% horse serum, 50 mM HEPES, 5 μg/mL Holo-transferrin, sodium selenite and 100 U/mL penicillin/streptomycin, at a pH of 7.4. Freshly egressed tachyzoites were used to infect myotubes with $7.2 \times 10^3$ tachyzoites per $cm^2$. Myotubes were washed in pre-warmed PBS 4 h after infection to remove not-invading parasites. The medium was replenished every 2–3 days.

### Reporting summary

Further information on research design is available in the Nature Portfolio Reporting Summary linked to this article.

## Data availability

All data are available within the paper and its supplementary information. Sequences used in this study have been obtained from VEuPathDB (https://veupathdb.org/) and ToxoDB (https://toxodb.org). All biological materials and data are available from the author upon request. Representative tomograms showing an apical end of ICMAP1-, ICMAP2-, ICMAP3[I]-, and ICMAP3[II]-depleted *T. gondii* are available in the Electron Microscopy Data Bank (EMDB) under accession codes EMD-42118, EMD-42119, EMD-42120, and EMD-42121, respectively. Source Data are provided with this paper.

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

## Acknowledgements

This work was supported by the Swiss National Foundation (grant no. 310030_185325) and by the European Research Council (ERC) under the European Union's Horizon 2020 research and innovation program agreement (grant no. 695596) to D.S.-F.; by a David and Lucile Packard Fellowship for Science and Engineering (2019–69645), a Burroughs Wellcome Fund Investigators in the Pathogenesis of Infectious Disease Program Award (1022785) and a Pennsylvania Department of Health FY19 Health Research Formula Fund to Y.-W.C.; by a Martin and Pamela Winter Infectious Disease Fellowship to M.M.; and by the National Institutes of Health (R01AI112427 and R01AI127798) to B.S.

We thank Dr Rouaa Ben Chaabene, Oksana Fiammingo and Alessandro Bonavoglia for their technical help, Dr Gaëlle Lentini for her insightful comments, as well as the team at the Bioimaging Core Facility (University of Geneva) for their technical assistance. We thank Dr Karine Frénal for sharing a detailed 'gliding assay' protocol. We thank Dr Stefan Steimle for his technical assistance with the Titan Krios G3i electron microscope.

## Author contributions

Conceptualization: N.D.S.P., A.G., Y.-W.C. and D.S.-F. Investigation: N.D.S.P., A.G., A.T.I.P., M.M., B.M., N.T., M.L. and E.D.-B. Data analysis: N.D.S.P., A.G., A.T.I.P., M.M., B.M., N.T. and M.L. Writing—original draft: N.D.S.P. and A.G. Writing—review and editing: N.D.S.P., A.G., B.S., Y.-W.C. and D.S.-F. Visualization: N.D.S.P., A.G., A.T.I.P., M.M., B.M., N.T. and E.D.-B. Supervision: B.S., Y.-W.C. and D.S.-F. Funding acquisition: Y.-W.C. and D.S.-F.

## Competing interests

The authors declare no competing interests.
