## [Peer Review File · Nature Communications]

Sustained rhoptry docking and discharge requires *Toxoplasma gondii* intraconoidal microtubule-associated proteinsREVIEWER COMMENTS

Reviewer #1 (Remarks to the Author):

In this paper the authors have tested the role of several proteins in the apical complex of *Toxoplasma* using a combination of genetic disruption and high resolution imaging. More specifically, they identified 2 intraconoidal microtubule associated proteins (IMCTs) and dissected their role in invasion of host cells by assessing their role in docking of rhoptries to the apical vesicle, which is required for rhoptry discharge. By deleting IMCTs, they show that a reduced number of rhoptries is enabled to link to Avs, reducing the ability to secrete rhoptry contents and attempt several invasion events. Overall this is an important and elegant study with some wider evolutionary implications on related pathogens with varied numbers of rhoptries.

In general the study is well performed, but there are a few things that should be clarified:

- 1) The authors see reduced plaque size but do not show plaque numbers. This should be done as a reduction in plaque size only can have various other roles while a reduction in plaque number would be more indicative of an invasion defect.
- 2) Integration PCRs are not looking at the WT locus of the gene as far as I can see. This should be added to ensure that no gene duplication could have occurred. This is in particular important because complementation of the transgenic lines (merodiploid lines for example to show that the deletion has no impact in the presence of a WT copy are absent throughout).
- 3) What is the size of ICMAP3?
- 4) There is no discussion on why ICMAP3 vanishes visually but not by Western?
- 5) Please add numbers of imaged cells for the manuscript.
- 6) Please comment on the differences in information you could obtain using Cryo-ET and FIB-SEM. This would be useful information for the field to decide on the best technique to investigate similar aspects of *Toxoplasma* biology.
- 7) Please add the cyst formation data mentioned in the discussion. No reason to not show it when discussed.
- 8) The mouse protection data is not sufficiently discussed. It is not clear how protection against challenge could be obtained without antibodies detected. This is specifically surprising since rhoptry contents are being secreted, just not as much. This would be a significant observation well beyond the parasitology field, and should be very thoroughly investigated. How many parasites have been injected (plaque assays to confirm parasite numbers)? Heat-killed parasites as a control to show that with low numbers of non-viable parasites there is no seroconversion etc.

Reviewer #2 (Remarks to the Author):

Nicolas Dos Santos Pacheco, Albert Tell i Puig, Amandine Guérin, Matthew Martinez, Bohumil Maco, Nicolò Tosetti, Matteo Lunghi, Boris Striepen, Yi-Wei Chang, Dominique Soldati-Favre
The intraconoidal microtubules orchestrate rhoptry discharge in *Toxoplasma gondii* to subvert the host

Context: Apicomplexan parasites are named for a set of unique secretory and cytoskeletal structures that are located at their apical end and are used to invade and egress from host cells. Some of the coccidian set of apicomplexans such as *Toxoplasma* have increased apical complex complexity. These parasites have a cone-shaped, tubulin filament containing structure termed the conoid. Within the lumen of the conoid are two closely adhered intraconoidal microtubules which are in close proximity to rhoptries and micronemes and an apical vesicle (AV) that docks one or two rhoptries to a macromolecular secretory apparatus. Although many apicomplexan parasites have 1 or 2 rhoptries, *Toxoplasma* has 10-12, making it capable of many rounds of secretion. This permits *Toxoplasma* to secrete into cells without invasion to introduce effectors into uninfected cells that

may modulate host responses.

Findings: This report describes two ICMAP associated proteins (ICMAP2 and ICMAP3), with ICMAP3 having two isoforms (A prior ICMT associated protein (ICMAP1) was characterized by Ke Hu's group). While ICMAP2 and the short ICMAP3 isoform (I) localize to the length of the ICMTs, the long isoform (II) of ICMAP3 localizes to a small punctum in the basal portion of the ICMTs. While ICMAP1, ICMAP2 and ICMAP3I are present in late-stage daughter buds, ICMAP3II is only detected in mature parasites. Induced depletion of ICMAP2 or ICMAP3I caused fitness defects manifested as reduced plaque size. While extracellular gliding, induced egress and microneme secretion are not affected, rhoptry secretion, invasion, and rhoptry docking are affected, as is virulence in a mouse model of infection.

Feedback: The images in this paper are outstanding, but the inclusion of negative "no effect" data obscures the impact of the significant findings, particularly for non-specialist readers. I strongly suggest that some of the very complex multi-panel figures be simplified by relegating "no effect" findings to supplemental data.

Specific changes recommended to streamline figures:

Figure 1: Panels C and D validation of the downregulation of ICMAP1-, ICMAP2-, ICMAP3I- and ICMAP3II-mAiD-HA by IFA and Western Blot should be in supplemental data.

Figure 4: a- ICMAP2- and ICMAP3I-depleted parasites are not impaired in gliding motility, egress, or microneme secretion (a, b, d,) and the negative data in panels f and g should be moved to supplemental data. This enhances the impact of the significant findings: invasion is impaired (c), rhoptry secretion is reduced (e) and many rhoptries are undocked (h).

New analysis: ICMAP2 and ICMAP3 were identified using the hyperLOPIT dataset and informatics approaches. While the manuscript text extensively describes how the authors determined that the TGME49_285140 locus is part of TGME49_285150, this content should be relegated to supplemental data. In its place, I strongly suggest that the authors analyze these proteins for conserved motifs. Although I was unable to easily access the authors' reannotation to obtain predicted amino acid sequences, I found some interesting characteristics that should be more extensively analyzed as part of a first paper on these proteins.

By my quick analysis with RADAR, TGME49_285140 has several highly repetitive sequences, which is often a hallmark of MAPs. For example, there are 14 copies of RREGEEERRRR or a close variant. It also has regions that BLAST identifies as similar to MAEBL.

TGME49_285150 does not have significant repeats, although there is some sequence noted. When the two sequences are stitched together these may become more significant. It also has a region that BLAST identifies as similar to MAEBL. Importantly, it has a potential IQ motif.

TGME49_224700 does not have significant repeats but has a motif similar to the phagosome trafficking protein DotA identified by Motif Finder.

By describing these features, the differences between ICMAP2 and the 2 forms of ICMAP3 will help convey how they are distinct. This information could be represented graphically in a panel that substitutes for one of the panels I suggest removing from Figure 1.

The title could be better: The current title doesn't reflect the full content of the paper ("The intraconoidal microtubules orchestrate rhoptry discharge in *Toxoplasma gondii* to subvert the host").

This title implies that you have evidence of specific processes beyond the findings presented here.

How about "Sustained rhoptry docking and discharge requires *Toxoplasma gondii* intraconoidal microtubule associated proteins"

Other experiments that I'd like to see but may be beyond the scope of this paper:

Expression of ICMAP2 and the 2 forms of ICMAP3 in fibroblast cells and/or microtubule pull-downs

with individual proteins to assess whether each directly binds microtubules or requires association with a second protein to localize. See ICMAP1 paper by Ke Hu.

Other information for the discussion: it might be worth noting the different types of host cells inhabited by *Toxoplasma* versus apicomplexans that live in RBCs or intestinal cells.

Reviewer #3 (Remarks to the Author):

In this paper the authors identify and characterize the localization and function of two ICMAP proteins in *T. gondii*. Using ultrastructure expansion microscopy the authors were able to characterize the localization of the ICMAPs relative to the ICMTs in developing intracellular parasites and extracellular tachyzoites at high resolutions. Knockdown of two ICMAP proteins resulted in defects in both invasion, rhoptry discharge as well as in host cell lysis. Cryo-ET characterization of the ICMAP knockdown tachyzoites revealed defects in the number of ICMTs, their length and in rhoptry docking. Experiments were thoughtfully designed, well executed and thoroughly analysed, resulting in a very interesting body of work that we very much enjoyed learning about. The manuscript was clear, logical and well written. Below, we outline a few points that we encountered when reviewing the manuscript, and hope that these notes are helpful for the authors.

Comments:

Figure S1C: the solubility experiment, methods and conclusions were somewhat confusing to us. If testing for solubility, one thing that was confusing to us is why the incubation was done on ice for 30 minutes? Additionally, proteins may end up in soluble/insoluble fractions for a variety of reasons, including aggregation, misfolding during handling, exact buffer compositions, etc. So it would be helpful if the authors could: 1) provide possible interpretations of these results, and 2) correlate the results with what might be expected for each protein - what can we learn from the AlphaFold predictions, and other bioinformatic predictions? Are the results consistent or inconsistent with this?

Figure S1F: The percentage of invasion looks pretty high, but the corresponding statement in the text suggests it is "modestly" impaired. It seems the difference is likely not statistically significant? This was a bit confusing for us.

Figure 2b: The representative data shown are beautiful. Could the authors please clarify how the analysis was carried out, number analyzed, how many measurements were made etc.?

Figure 2D: What is the reason that ICMAP2 deletion strain results in partial loss of ICMAP1? This was not immediately clear to us.

Figure 2G/H: ICMAP3' is still robustly expressed and present on the western. Although it indeed seems mislocalized/ not localized to the apex, where is it localized to? We found it hard to observe the signal for it.

Is ICMAP2 required for localization or stability of ICMAP3, or could it be both? Do the data speak to this? If not, it would be helpful for the conclusion to explore different possibilities.

Figure 3: in the ICMAP1 KD tomograms, the shape of both the micronemes and the MVs appear to be different from the WT parasites, is this expected?

Figure S4A: it may be helpful to add data points, if they are not too distracting

Figure 1: Please include scale bars in all the insets

Figure 2a, e j, g: Please include scale bars for the main micrographs

The cryo-ET data is of high quality, and is used in a very nice way that clearly answers a question.

Line 267/268: could the authors please indicate how many tomograms were analyzed, to give a sense to the reader of the sample size and heterogeneity

In some cases, the segmentation analysis does not accurately reflect the data. For example, Fig. 3b - in the 2D slice the MVs seem to line up but in 3D they don't (see screenshot from movie below). Also, MV's appear very circular in the reconstructions but the structure itself appears ovoid in the tomogram. Could the authors please review the segmentation analysis, and

ensure that the segmentation and 3D reconstructions accurately represent the tomograms.

Cryo-ET Methods: In the cryo-ET methods section, could the authors please include more details, for example, currently missing are: pixel size, number of tilt series acquired, details for freezing of cryo-ET grids (what instrument was used, and parameters for freezing), preprocessing steps such as motion correction.

In figures, could the authors please show un-annotated and annotated tomogram slices side-by-side, so that the reader can see the data, and understand the annotation better.

Currently un-annotated slices are in the supplementary material, but it would be preferable to show this in the main text.

Please deposit cryo-ET data in EMPIAR, and provide the deposition code in data availability statement

We noticed that the microneme shape is quite variable across tomograms. Do the authors have thoughts on why this may be? This is just a curiosity question.

It would be helpful for the authors to more clearly state when going between intracellular and extracellular stages, especially for readers who may be interested but not in the toxoplasma field

Reviewer #1

In this paper the authors have tested the role of several proteins in the apical complex of *Toxoplasma* using a combination of genetic disruption and high resolution imaging. More specifically, they identified 2 intraconoidal microtubule associated proteins (IMCTs) and dissected their role in invasion of host cells by assessing their role in docking of rhoptries to the apical vesicle, which is required for rhoptry discharge. By deleting IMCTs, they show that a reduced number of rhoptries is enabled to link to Avs, reducing the ability to secrete rhoptry contents and attempt several invasion events. Overall this is an important and elegant study with some wider evolutionary implications on related pathogens with varied numbers of rhoptries. In general the study is well performed, but there are a few things that should be clarified:

1) The authors see reduced plaque size but do not show plaque numbers. This should be done as a reduction in plaque size only can have various other roles while a reduction in plaque number would be more indicative of an invasion defect.

The protocol used for plaque assay does not involve the precise counting of input parasites and its purpose is not to assess the number of plaques but only the plaque size. This assay measures the overall fitness of a given parasite strain during multiple lytic cycles. It is correct that a decrease in plaque size could reflect a defect in one or more of the multiple steps of the lytic cycle, including intracellular growth, egress, gliding and invasion. The plaque assay is not intended to be used as an invasion assay. Instead, a tailored invasion assay is used that allows a robust comparative and quantitative analysis. Importantly, this invasion assay distinguishes fluctuation of viability in a population of parasites from impairment in invasion which is not possible to assess in a plaque assay. Accordingly, we performed specific tests such as an invasion assay for the fitness conferring strains, here ICMAP2 and ICMAP3^I (see original figure 4C) and non-fitness conferring strains like ICMAP1 and ICMAP3^{II} (see original supplementary figure 1F). Upon revision, those assays are now presented in the **new figure 4A and supplementary figure 1F**.

2) Integration PCRs are not looking at the WT locus of the gene as far as I can see. This should be added to ensure that no gene duplication could have occurred. This is in particular important because complementation of the transgenic lines (merodiploid lines for example to show that the deletion has no impact in the presence of a WT copy are absent throughout).

We agree that the PCR analysis to assess integration that was performed in the original submission was minimal. More extensive integration analyses of the mutant strains have been performed. For each genomic PCR analysis, a scheme is presented along the results for clarity and the results are presented in the **new supplementary figure 1B, 1G, 2D, 2E and 6C**.

3) What is the size of ICMAP3?

From our analysis of the ICMAP3 locus we can only assess the size of the short product ICMAP3^I. The size of the protein is predicted around 47 kDa, as shown in the original figure S1A. Concerning the long isoform ICMAP3^{II}, we cannot precisely determine the size due to the two long stretches of sequence repeats. Those repeats prevent cDNA amplification, precise determination of the boundaries of the introns and accurate gene annotation. We can only deduce it from the predicted ToxoDB model to be around 600 kDa. We have only confirmed experimentally the first four exons (we amplified those by PCR and sequenced the product – see original figure S2B), and the STOP codon.

4) There is no discussion on my ICMAP3 vanishes visually but not by Western?

Indeed, depletion of ICMAP2, and therefore ICMTs destabilization, prevent ICMAP3^I apical accumulation without affecting the protein level when assessed by western blot (see original figure 2G-H). We postulate that ICMAP2 is required for the recruitment of ICMAP3^I at the ICMTs but not for the stability of the protein. The loss of apical signal is plausibly the result of a dispersion of ICMAP3^I in the cytosol, which gets diluted and not detectable anymore by IFA.

5) Please add numbers of imaged cells for the manuscript.

For the quantitative data, such as cryo-ET, the total number of cells is included in the figure legend. For full transparency and convenience, we are also including with this revised submission a **source data file** (Excel file) compiling all the numerical data of the manuscript.

6) Please comment on the differences in information you could obtain using Cryo-ET and FIB-SEM. This would be useful information for the field to decide on the best technique to investigate similar aspects of *Toxoplasma* biology.

The FIB-SEM method used in this manuscript involves fixation and staining of the sample, which disrupt some cellular structures, involves imaging at a lower magnification, thus resulting in lower resolution but offers nice contrast of membranes and large structures. Cryo-ET preserves cellular structure since it does not involve fixation and staining and imaging is taken at higher magnification. Therefore, native cellular features, such as smaller structures (like ICMTs, MVs, AV) and protein structures through sub-tomogram averaging (not done in this paper) can be observed. In addition, cryo-ET can be performed in a more quantitative manner when sufficient cells are imaged and analyzed, something more difficult to perform using FIB-SEM which is a very demanding method. Few sentences about the main differences have been added to the revised manuscript in the **discussion section (see line 332-336)** and read as follows:

“Our study used two complementary techniques, FIB-SEM and cryo-ET, to examine the role of the ICMAPs. FIB-SEM is performed on fixed samples at a comparatively low resolution but allows a greater membrane contrast of large structures. Cryo-ET on the other hand is performed on native samples preserving cellular structures and at higher resolution allowing the analysis of smaller structures.”

7) Please add the cyst formation data mentioned in the discussion. No reason to not show it when discussed.

As rightfully requested, the data concerning the cyst formation has been added to the revised manuscript in the **new figure 7E and supplementary figure 6C-D-E-F**. Briefly, we show that Δ ICMAP2 parasites generated in the ME49 background are able to express bradyzoites-specific markers and form cyst in myotubes. We also show that like their counterpart of the RH background, they are unable to elicit seroconversion upon mice injection.

8) The mouse protection data is not sufficiently discussed. It is not clear how protection against challenge could be obtained without antibodies detected. This is specifically surprising since rhoptry contents are being secreted, just not as much. This would be a significant observation well beyond the parasitology field, and should be very thoroughly investigated. How many parasites have been injected (plaque assays to confirm parasite numbers)? Heat-killed parasites as a control to show that with low numbers of non-viable parasites there is no seroconversion etc.

We thank the reviewer for this comment. Our data show that Δ ICMAP2 parasites are unable to generate an infection and are probably cleared before appearance of anti-*Toxoplasma* antibodies in the mice. Hence the initial survival of the mice after the first infection by Δ ICMAP2 parasites. Logically, a challenge with Δ ICMAP2 parasites, even at high dose, was again not sufficient to generate an infection and a seroconversion. We think that if the mice that did not seroconvert after the initial Δ ICMAP2 infection were challenged with wild-type parasite, they would have likely succumbed due to the infection: those mice are not protected.

The revised manuscript has been modified to make the **discussion section** clearer on this topic (**see line 397-401**) and reads as follow:

“Mouse infection conducted here revealed that parasites mutants lacking ICMAP2 are still able to invade tissue culture but are completely avirulent in vivo. Remarkably, this mutant failed to trigger seroconversion in vivo, suggesting a fast clearance of those parasites by the innate immune system.”

Concerning the number of parasites injected during the initial infection, we indeed performed a plaque assay as a control. The results show that the infection has been carried out in a satisfactory manner. The results are as follows:

- RH parental (100 parasites) = 9 plaques after 1 week
- Δ ICMAP2 (100 parasites) = 4 plaques after 1 week
- Δ ICMAP2 (1'000 parasites) was not analyzed by plaque assay

Reviewer #2

Context: Apicomplexan parasites are named for a set of unique secretory and cytoskeletal structures that are located at their apical end and are used to invade and egress from host cells. Some of the coccidian set of apicomplexans such as *Toxoplasma* have increased apical complex complexity. These parasites have a cone-shaped, tubulin filament containing structure termed the conoid. Within the lumen of the conoid are two closely adhered intraconoid microtubules which are in close proximity to rhoptries and micronemes and an apical vesicle (AV) that docks one or two rhoptries to a macromolecular secretory apparatus. Although many apicomplexan parasites have 1 or 2 rhoptries, *Toxoplasma* has 10-12, making it capable of many rounds of secretion. This permits *Toxoplasma* to secrete into cells without invasion to introduce effectors into uninfected cells that may modulate host responses.

Findings: This report describes two ICMAP associated proteins (ICMAP2 and ICMAP3), with ICMAP3 having two isoforms (A prior ICMT associated protein (ICMAP1) was characterized by Ke Hu's group). While ICMAP2 and the short ICMAP3 isoform (I) localize to the length of the ICMTs, the long isoform (II) of ICMAP3 localizes to a small punctum in the basal portion of the ICMTs. While ICMAP1, ICMAP2 and ICMAP3I are present in late-stage daughter buds, ICMAP3II is only detected in mature parasites. Induced depletion of ICMAP2 or ICMAP3I caused fitness defects manifested as reduced plaque size. While extracellular gliding, induced egress and microneme secretion are not affected, rhoptry secretion, invasion, and rhoptry docking are affected, as is virulence in a mouse model of infection.

Feedback: The images in this paper are outstanding, but the inclusion of negative "no effect" data obscures the impact of the significant findings, particularly for non-specialist readers. I strongly suggest that some of the very complex multi-panel figures be simplified by relegating "no effect" findings to supplemental data.

Specific changes recommended to streamline figures:

Figure 1: Panels C and D validation of the downregulation of ICMAP1-, ICMAP2-, ICMAP3I- and ICMAP3II-mAiD-HA by IFA and Western Blot should be in supplemental data.

Following the reviewer's recommendation, several figures have been streamlined in the revised manuscript to enhance clarity. For the figure 1 specifically, we moved the original IFA panel to the **new supplementary figure 1E**. We chose to keep the western blot panel in the main figure (see **figure 1D**) as it is not only informative for the depletion of the proteins, but also for the assessment of their sizes, which we consider relevant.

Figure 4: a- ICMAP2- and ICMAP3I-depleted parasites are not impaired in gliding motility, egress, or microneme secretion (a, b, d,) and the negative data in panels f and g should be moved to supplemental data. This enhances the impact of the significant findings: invasion is impaired (c), rhoptry secretion is reduced (e) and many rhoptries are undocked (h).

Here as well, we streamlined the revised figure 4 according to the reviewer's recommendation. The gliding motility, egress and microneme secretion panels have been moved to the **new supplementary figure 4B-C-D**. In addition we moved the IFA concerning Nd6 stability in the **new supplementary figure 4F**.

We chose to keep the U-ExM panel concerning the Nd6 stability in the main figure (**new Figure 4C**) as it is a conceptually important result: the RSA and rosette are not affected by ICMTs destabilization. Therefore, their biogenesis/formation are still an open and important question. Also, U-ExM allows here a clear observation of the RSA position at an unprecedented resolution using light microscopy.

New analysis: ICMAP2 and ICMAP3 were identified using the hyperLOPIT dataset and informatics approaches. While the manuscript text extensively describes how the authors determined that the TGME49_285140 locus is part of TGME49_285150, this content should be relegated to supplemental data. In its place, I strongly suggest that the authors analyze these proteins for conserved motifs. Although I was unable to easily access the authors' reannotation to obtain predicted amino acid sequences, I found some interesting characteristics that should be more extensively analyzed as part of a first paper on these proteins.

By my quick analysis with RADAR, TGME49_285140 has several highly repetitive sequences, which is often a hallmark of MAPs. For example, there are 14 copies of RREGEEERRRR or a close variant. It also has regions that BLAST identifies as similar to MAEBL.

TGME49_285150 does not have significant repeats, although there is some sequence noted. When the two sequences are stitched together these may become more significant. It also has a region that BLAST identifies as similar to MAEBL. Importantly, it has a potential IQ motif.

TGME49_224700 does not have significant repeats but has a motif similar to the phagosome trafficking protein DotA identified by Motif Finder.

By describing these features, the differences between ICMAP2 and the 2 forms of ICMAP3 will help convey how they are distinct. This information could be represented graphically in a panel that substitutes for one of the panels I suggest removing from Figure 1.

As suggested, the paragraph describing the analysis of the ICMAP3 locus has been removed from the main text and instead placed as a **supplementary discussion**.

As requested, we added schematics of the 4 proteins studied in in the **revised figure 1B**. We agree with the presence of an IQ motif in ICMAP3^I and an MAEBL related sequence in ICMAP3^{II}. However, analysis of the ICMAP2 protein sequence only gives poor confidence score concerning the DotA motif (e-value = 0.4 with MotifFinder) and we decided not to include it to the scheme.

Concerning the repeats in the ICMAP3 ORF, we also noticed those intriguing repeats initially. However, we did not emphasize on their presence due to the uncertainty in the gene annotation. When BLASTed, those repeats do not seem to resemble any known repeated sequences. We decided to include a logo of those two repeats in the schemes of the **revised figure 1B**.

The title could be better: The current title doesn't reflect the full content of the paper ("The intraconoidal microtubules orchestrate rhoptry discharge in *Toxoplasma gondii* to subvert the host"). This title implies that you have evidence of specific processes beyond the findings presented here. How about "Sustained rhoptry docking and discharge requires *Toxoplasma gondii* intraconoidal microtubule associated proteins".

We thank the reviewer for the suggestion. The title of the study is now "**Sustained rhoptry docking and discharge requires *Toxoplasma gondii* intraconoidal microtubule-associated proteins.**"

Other experiments that I'd like to see but may be beyond the scope of this paper: Expression of ICMAP2 and the 2 forms of ICMAP3 in fibroblast cells and/or microtubule pull-downs with individual proteins to assess whether each directly binds microtubules or requires association with a second protein to localize. See ICMAP1 paper by Ke Hu.

We also thought about such experiments to be included in the manuscript. Especially microtubule binding assay and expression of the proteins in mammalian cells (xenopus cells like the original ICMAP1 study). However, recombinant expression of ICMAP2 was not soluble even in insect cells and ICMAP3 was not attempted due to its intimidating large size.

Other information for the discussion: it might be worth noting the different types of host cells inhabited by *Toxoplasma* versus apicomplexans that live in RBCs or intestinal cells.

The apicomplexans living in red blood cells such as *Plasmodium* merozoites only possess two rhoptries. Logically, this parasite does not need to establish an environmental niche by "pocking" surrounding cells with rhoptries without infecting them. *Cryptosporidium* possesses a single rhoptry but again, its infection rapidly leads to a devastating destruction of the intestinal epithelium, without the need of establishing a long-term "niche". In contrast, *Toxoplasma* goes in deep tissues (muscle, brain, etc.) to establish a long-term infection and hence needs to modulate the environment and the immune response to control the infection, to prevent death of its host. This could explain the presence of such an intricate iterative rhoptry discharge system in *Toxoplasma*.

The following sentence have been added to the **revised discussion section (see line 383-390)**:

"Apicomplexans such as Plasmodium and Cryptosporidium might not need to subvert neighboring host cells through iterative rhoptry discharge. While Plasmodium circulates continuously in the blood stream and only modifies the infected cells to limit splenic clearance, Cryptosporidium infection leads to rapid destruction of the intestinal epithelium and shedding in the environment. Contrastingly, Toxoplasma infect deep tissues to establish a chronic infection and requires the modulation of the environment and the immune response to avoid killing the host [55]. This in turn could explain why iterative rhoptry discharge is only found in Toxoplasma and parasites with similar lifestyle."

Reviewer #3

In this paper the authors identify and characterize the localization and function of two ICMAP proteins in *T. gondii*. Using ultrastructure expansion microscopy the authors were able to characterize the localization of the ICMAPs relative to the ICMTs in developing intracellular parasites and extracellular tachyzoites at high resolutions. Knockdown of two ICMAP proteins resulted in defects in both invasion, rhoptry discharge as well as in host cell lysis. Cyro-ET characterization of the ICMAP knockdown tachyzoites revealed defects in the number of ICMTs, their length and in rhoptry docking. Experiments were thoughtfully designed, well executed and thoroughly analysed, resulting in a very interesting body of work that we very much enjoyed learning about. The manuscript was clear, logical and well written. Below, we outline a few points that we encountered when reviewing the manuscript, and hope that these notes are helpful for the authors.

Comments:

Figure S1C: the solubility experiment, methods and conclusions were somewhat confusing to us. If testing for solubility, one thing that was confusing to us is why the incubation was done on ice for 30 minutes? Additionally, proteins may end up in soluble/insoluble fractions for a variety of reasons, including aggregation, misfolding during handling, exact buffer compositions, etc. So it would be helpful if the authors could: 1) provide possible interpretations of these results, and 2) correlate the results with what might be expected for each protein - what can we learn from the

AlphaFold predictions, and other bioinformatic predictions? Are the results consistent or inconsistent with this?

The 30 min incubation on ice might not be critical for this assay. However, it ensures the depolymerization of potentially contaminating host cell microtubules.

In the solubility experiment, we show that ICMAP1 and ICMAP3¹ are solubilized while ICMAP2 needs harsh conditions to be extracted. Given the ICMTs localization of the three proteins, this experiment gives initial clues about their biochemical properties: ICMAP2 is not readily soluble and might be tightly bound to the ICMTs, while ICMAP1 and ICMAP3¹ seem more loosely attached. For ICMAP1 this result is interesting in the context of its localization to the fibers or its ability to form fibers. This awaits further investigations. ICMAP3¹ disappears when ICMAP2 is depleted, whereas when ICMAP3¹ is depleted, ICMAP2 is still apical, again suggesting strong binding to the ICMTs. For those reasons we think that the results regarding solubility are overall consistent with the other data presented.

We have run AlphaFold predictions for the 4 proteins and the results were not compelling with mostly unstructured “spaghetti bowl” predictions, from which nothing could be concluded.

Of note, we realized that the fractionation assay was not performed on the ICMAP3¹-mAiD-HA strain as initially and mistakenly stated but a ICMAP3¹-Ty strain instead since the Ty antibodies are cleaner. The **revised supplementary figure 1C and its associated legend** have been corrected accordingly.

Figure S1F: The percentage of invasion looks pretty high, but the corresponding statement in the text suggests it is “modestly” impaired. It seems the difference is likely not statistically significant? This was a bit confusing for us.

For ICMAP1 depletion, a ~10% decrease, from 74% to 63%, in invasion is observed (**see new supplementary figure 1F**). This modest 10% decrease is not apparent in the plaque assay and is consistent with the invasion assay being more precise. However, thanks to the comment of the reviewer we realized that our statistical tests for this assay were not optimal: they were comparing ICMAP1-IAA with ICMAP1+IAA, while a more correct test is to compare the Tir1+IAA with the ICMAP1+IAA (as it is done elsewhere in the study). When those conditions are compared, this ~10% decrease in invasion is not statistically significant. The **text (line 128-129)** and the **new supplementary figure 1F** have been corrected accordingly.

Figure 2b: The representative data shown are beautiful. Could the authors please clarify how the analysis was carried out, number analyzed, how many measurements were made etc.?

Only one picture (the one presented) was carefully measured and analyzed. The whole analysis was performed using the LAS X software (Leica microscope) and its built-in “line profile” function. In short, we are able on the maximum projection image to trace a line on the image (**see figure 2B or 2F**). The intensity of the different signal along this line is automatically transformed into numerical data that can be imported in an Excel or GraphPad file for plotting (**see new source data**). This protocol is succinctly described in the **method section (line 524-526)**.

Those ICMAP1-ICMAP2 shifted signals were clearly observed in many parasites, enough for us to notice it immediately at the microscope. Unfortunately, noticing this phenomenon and capturing it in a clear image was not as straightforward. Also it is not that easy to get parasites with a conoid nicely extruded, a clean tubulin staining, and clean HA and Ty staining. Also depending on the parasite orientation, the shift in the two signals is not always evident on maximum projected images.

The main message here is that the offset signal was a clear characteristic of ICMAP1. While we analyzed only one picture, this analysis is part of a body of evidence all pointing in the same direction:

- 1- IFA showed a non-perfect colocalization between ICMAP1 and ICMAP2.
- 2- U-ExM confirmed these results by showing that ICMAP1 signal is more apical and offset to one side suggesting an association with the ICMTs fibers.
- 3- Cryo-ET confirmed the role of ICMAP1 in the formation/stability of those fibers.

Figure. 2D: What is the reason that ICMAP2 deletion strain results in partial loss of ICMAP1? This was not immediately clear to us.

While ICMAP2 depletion leads to a clear disorganization of the ICMTs, 80% of the parasites analyzed by cryo-ET (see revised Figure 5F) retained at least one ICMT inside the conoid. Therefore, we think that the ICMAP1 signal still visible in ICMAP2-depleted parasites could be due to the ICMTs still inside the conoid, and probably with intact fibers.

Figure 2G/H: ICMAP3¹ is still robustly expressed and present on the western. Although it indeed seems mislocalized/ not localized to the apex, where is it localized to? We found it hard to observe the signal for it.

Indeed we do not observe a clear signal for ICMAP3¹ by IFA upon ICMAP2 depletion. We suspect that the signal might simply be diluted in the cytosol upon ICMTs destabilization.

Is ICMAP2 required for localization or stability of ICMAP3, or could it be both? Do the data speak to this? If not, it would be helpful for the conclusion to explore different possibilities.

We showed in the original figure 2I, that the depletion of ICMAP3¹ does not perturb the localization of ICMAP2. Hence it seems that ICMAP2 is more at the roots of the observed phenotypes. In other words, when ICMAP3¹ is depleted, the phenotype results from the absence of the ICMAP3¹ only. Contrastingly, when ICMAP2 is depleted, the phenotype observed is due to the absence of both ICMAP2 and ICMAP3¹. We could draw a parallel with the solubility results in which ICMAP2 is insoluble while ICMAP3¹ is mostly soluble. This speaks for ICMAP2 being strongly attached to the ICMTs while ICMAP3¹ might solely be associated with other MAPs including ICMAP2 or be loosely attached directly to the ICMTs.

Figure 3: in the ICMAP1 KD tomograms, the shape of both the micronemes and the MVs appear to be different from the WT parasites, is this expected?

Indeed some variation of the shape of micronemes and MVs has been observed in mutant parasites but also in WT parasites. We postulate that it could be a consequence of the loss of plasma membrane integrity during blotting, however because the imaging can only focus on the apical region of the parasite (that is thin enough to acquire good resolution) we cannot establish a firm correlation between the two phenomena. Importantly ICMAP1 depletion leads to no defect of microneme secretion.

Figure S4A: it may be helpful to add data points, if they are not too distracting

We made a conscious exception for the intracellular growth graph to not show the data points of the individual replicates. First it makes the graph overly heavy. Also, those individual data points would be difficult to interpret as the data "2-4-8-16+" are stacked on top of one another. We think that visually the graph is clearer as is and opted not to modify it. Importantly, the raw data are available in the **source data file**.

Figure 1: Please include scale bars in all the insets.

Scale bars have been added to the insets as requested (see new figure 1C).

Figure 2a, e j, g: Please include scale bars for the main micrographs

Scale bars have been added to the main micrographs of the figure 2A-E-J-G as requested.

The cryo-ET data is of high quality, and is used in a very nice way that clearly answers a question. Line

267/268: could the authors please indicate how many tomograms were analyzed, to give a sense to the reader of the sample size and heterogeneity.

The number of tomograms analyzed are indicated in the legends of each relevant figure (20 for the WT and ICMAP1/ICMAP2 mutants and 10 for each ICMAP3 mutants) so a total of 80 tomograms across the different strains. This information is also included in **materials and methods (line 562-564)** to help the reader. The numerical data can now also be found in the **source data file**.

In some cases, the segmentation analysis does not accurately reflect the data. For example, Fig. 3b - in the 2D slice the MVs seem to line up but in 3D they don't (see screenshot from movie below). Also, MV's appear very circular in the reconstructions but the structure itself appears ovoid in the tomogram. Could the authors please review the segmentation analysis, and ensure that the segmentation and 3D reconstructions accurately represent the tomograms.

As the reviewer pointed out, some MVs do have slightly inconsistent size and shape. We have observed this phenomenon in both the wild-type and different ICMAP mutant cells but could not correlate any of such variations with particular phenotype or strains. Therefore, we opted to represent the MVs (and the AV) in 3D segmentations as simple spherical objects to focus the illustration on their presence and spatial distribution for display purpose. This is now stated in the **materials and methods (line 564-567)**. We note that all statistical measurements of cryo-ET data in this study (e.g., MV-ICMT2 and MV-MV distances; **supplementary figure 5**) were made on the raw densities of tomograms where we accurately select molecular features in 3D. In fact, such irregularity of vesicle shape and size has been faithfully indicated by the color overlay on all 2D tomographic slices shown (e.g., **figure 3A,B; 5C,H; 6C**). We also provide non-colored tomographic slices (**supplementary figure 7**) and representative tomograms of each condition (**EMDB; data availability statement**) for reader's inspection.

Cryo-ET Methods: In the cryo-ET methods section, could the authors please include more details, for example, currently missing are: pixel size, number of tilt series acquired, details for freezing of cryo-ET grids (what instrument was used, and parameters for freezing), preprocessing steps such as motion correction.

The number of acquired tilt-series, thus tomogram analyzed, is indicated in the **materials and methods section (line 562-564)**. We have now also added in the revised manuscript additional information on the instrument and freezing parameters as well as the pixel size with the preprocessing steps. The added information is underlined:

Line 547-550: *"Briefly, 4 μ L containing around 4 million tachyzoites mixed with 10 nm gold fiducials were loaded onto an EM grid for plunge freezing into a liquid ethane/propane mixture on an EM GP2 automatic plunger (Leica Microsystems, Wetzlar, Germany) after 4 seconds of back blotting."*

Line 554-558: *"Tilt-series were collected with a span of 120° (-60° to +60°; bi-directional scheme) with 2° increments at a magnification of 33,000x, a nominal pixel size of 2.65 Å, and a defocus range of -1 to -4 μ m. Motion correction of images was done using the Alignframe function in IMOD."*

In figures, could the authors please show un-annotated and annotated tomogram slices side-by-side, so that the reader can see the data, and understand the annotation better. Currently un-annotated slices are in the supplementary material, but it would be preferable to show this in the main text.

Due to the lack of space in the main figures, we decided to only show annotated tomograms in the main figure. The addition of un-annotated tomograms would force us to display smaller images, impeding on the clarity of the figure. Instead, we now display the un-annotated and annotated tomograms side-by-side in the **revised supplementary figure 7** to help the reader understand the annotation better, based on the reviewer's great suggestion.

Please deposit cryo-ET data in EMPIAR, and provide the deposition code in data availability statement. As requested, we have deposited representative tomograms of each mutant displayed in this manuscript in the Electron Microscopy Data Bank (EMDB). This is now indicated in the **data availability statement (line 721-726)** that reads as follow:

“Representative tomograms showing an apical end of ICMAP1-, ICMAP2-, ICMAP3^l-, and ICMAP3^{ll}-depleted T. gondii are available in the Electron Microscopy Data Bank (EMDB) under accession codes EMD-42118 [<https://www.ebi.ac.uk/pdbe/entry/emdb/EMD-42118>], EMD-42119 [<https://www.ebi.ac.uk/pdbe/entry/emdb/EMD-42119>], EMD-42120 [<https://www.ebi.ac.uk/pdbe/entry/emdb/EMD-42120>], and EMD-42121 [<https://www.ebi.ac.uk/pdbe/entry/emdb/EMD-42121>], respectively.”

We noticed that the microneme shape is quite variable across tomograms. Do the authors have thoughts on why this may be? This is just a curiosity question.

As mentioned above in one of our previous answers to this reviewer for figure 3, we hypothesize that this could be a consequence of loss of plasma membrane integrity during sample blotting. However, because we are focusing cryo-ET imaging on the apical part (that is thin enough to achieve a good resolution) we cannot comprehensively examine the loss of the plasma membrane integrity throughout the entire parasite and correlate it with microneme shape variations among cells.

It would be helpful for the authors to more clearly state when going between intracellular and extracellular stages, especially for readers who may be interested but not in the toxoplasma field.

We have modified the figure legends of IFAs and U-ExM when necessary to indicate if the images displayed are from intracellular or extracellular parasites.

REVIEWERS' COMMENTS

Reviewer #1 (Remarks to the Author):

I am happy with the responses by the authors and would like to congratulate the team on an important study.

Reviewer #2 (Remarks to the Author):

I am satisfied with the revisions to the MS and recommend it for publication.

Reviewer #4 (Remarks to the Author):

We thank the authors for their thoughtful and thorough response to our comments, which have all been well addressed.

A minor comment that could be helpful to address prior to publication:

Figure S1C: Could the authors note that proteins can end up in the soluble and insoluble fractions due to aggregation or protein misfolding during handling.

Second round of revision:

Reviewer #1 (Remarks to the Author):

I am happy with the responses by the authors and would like to congratulate the team on an important study.

We would like to thank Reviewer #1 for his time and thoughtful comments regarding our study. We feel that, thanks to the suggestions made by all three reviewers, our manuscript gained in clarity and impact.

Reviewer #2 (Remarks to the Author):

I am satisfied with the revisions to the MS and recommend it for publication.

We would like to thank Reviewer #2 for his time and comments during the review process.

Reviewer #3 (Remarks to the Author):

We thank the authors for their thoughtful and thorough response to our comments, which have all been well addressed.

We would like to thank Reviewer #3 (and his team) for the thorough review of our manuscript.

A minor comment that could be helpful to address prior to publication:

Figure S1C: Could the authors note that proteins can end up in the soluble and insoluble fractions due to aggregation or protein misfolding during handling.

We added in the figure S1C legend a statement indicating that “proteins can end up in the soluble and insoluble fractions due to aggregation or protein misfolding during handling” as requested by the reviewer.